# Revealed versus potential spatial accessibility of healthcare and changing patterns during the COVID-19 pandemic

Kristina Gligorić [1,2,6], Chaitanya Kamath[1,6], Daniel J. Weiss [3,4,6], Shailesh Bavadekar[1], Yun Liu [1], Tomer Shekel[1], Kevin Schulman[5,7] & Evgeniy Gabrilovich [1,7✉]

## Abstract

**Background** Timely access to healthcare is essential but measuring access is challenging. Prior research focused on analyzing potential travel times to healthcare under optimal mobility scenarios that do not incorporate direct observations of human mobility, potentially underestimating the barriers to receiving care for many populations.

**Methods** We introduce an approach for measuring accessibility by utilizing travel times to healthcare facilities from aggregated and anonymized smartphone Location History data. We measure these revealed travel times to healthcare facilities in over 100 countries and juxtapose our findings with potential (optimal) travel times estimated using Google Maps directions. We then quantify changes in revealed accessibility associated with the COVID-19 pandemic.

**Results** We find that revealed travel time differs substantially from potential travel time; in all but 4 countries this difference exceeds 30 minutes, and in 49 countries it exceeds 60 minutes. Substantial variation in revealed healthcare accessibility is observed and correlates with life expectancy ($\rho=-0.70$) and infant mortality ($\rho=0.59$), with this association remaining significant after adjusting for potential accessibility and wealth. The COVID-19 pandemic altered the patterns of healthcare access, especially for populations dependent on public transportation.

**Conclusions** Our metrics based on empirical data indicate that revealed travel times exceed potential travel times in many regions. During COVID-19, inequitable accessibility was exacerbated. In conjunction with other relevant data, these findings provide a resource to help public health policymakers identify underserved populations and promote health equity by formulating policies and directing resources towards areas and populations most in need.

## Plain language summary

Spatial access to healthcare facilities (i.e., how long people need to travel to reach care) is important for understanding public health, but hard to measure. Most research so far has focused on theoretical (potential) travel times. Using anonymized smartphone location history data, we measure actual (revealed) travel times to healthcare facilities in over 100 countries. We find that revealed travel times exceed theoretical travel times in many regions of the world, meaning that in reality people travel longer to get healthcare. Our data also show that inequities in travel time became worse during the COVID-19 pandemic. When combined with other data, these results can help policymakers identify areas and populations at need, and direct resources to improve public health.

[1] Google Research, Mountain View, CA, USA. [2] Computer Science Department, Stanford University, Stanford, CA, USA. [3] Telethon Kids Institute, Perth Children's Hospital, Nedlands, WA, Australia. [4] Faculty of Health Sciences, Curtin University, Bentley, WA, Australia. [5] Clinical Excellence Research Center, School of Medicine and Graduate School of Business, Stanford University, Stanford, CA, USA. [6] These authors contributed equally: Kristina Gligorić, Chaitanya Kamath, Daniel J. Weiss. [7] These authors jointly supervised this work: Kevin Schulman, Evgeniy Gabrilovich. ✉email: gabr@acm.org

Healthcare accessibility is a multifaceted concept and a focal point in the United Nations' Sustainable Development Goals[1–3]. Yet, numerous global and country-specific analyses have confirmed that equitable access to healthcare remains aspirational[4–7]. More recently, the COVID-19 pandemic has challenged the capacity of healthcare systems across the globe, and along with travel restrictions, potentially creating new barriers to healthcare. However, few systematic efforts have characterized healthcare accessibility at the global scale and none have empirically assessed how this key public health metric has been impacted by the pandemic.

Barriers in access to healthcare increase the likelihood that needed healthcare will be delayed or foregone, thereby increasing patients' risks for chronic and acute conditions. Such barriers are particularly important for vulnerable populations with lower incomes or the under-insured and uninsured, and substantial differences in healthcare access exist between rural and urban areas[8–11]. Examples illustrating the relationship between healthcare access and outcomes include studies exploring the prevalence of diabetes and related adverse outcomes[12]; low rates of prenatal care and the associated increased risk of having premature or low-birth-weight infants[13,14]; breast cancer screening practices[15]; and delayed care-seeking and timely diagnosis of appendicitis rupture[16]. Access barriers among patients also affect the quality of clinical encounters as they lead to decreased physician trust and disparities in subsequent outcomes[17,18].

A key factor associated with healthcare accessibility is how long people have to travel to reach healthcare providers. While the concept of spatial access is intuitive, measuring this dimension of healthcare systems is challenging. However, modern computational tools and human mobility datasets provide an unparalleled opportunity for characterizing healthcare accessibility. Approaches for measuring spatial accessibility to healthcare are based on potential and revealed accessibility indicators[4,7,19]. Potential accessibility captures the opportunities available to the population and is a function of the proximity to the providers, state of the transportation routes, and access to public or private transportation. Revealed accessibility, in contrast, measures the actual efforts to reach a healthcare facility by a given population, as estimated from patient lists, records of actual travel behavior, or as reported in surveys[1,2]. Whereas potential (theoretical) accessibility based on routes and population density is helpful in planning, it is important to understand if reality (revealed accessibility) deviates from theory, and by how much. Understanding this difference and how it is influenced by crises (such as the COVID-19 pandemic) can help further refine planning and catalyze efforts to improve healthcare accessibility.

This paper reports an approach to the development of measures of both potential and revealed access to care in over 100 countries. The inventory of geolocated hospitals and medical centers providing urgent and emergency care was extracted from Google's Maps and Search datasets (Methods). We estimated potential accessibility using Google Maps Platform Directions API, the same API that powers navigation in Google Maps[20]. To produce revealed spatial accessibility maps, we leveraged the spatial patterns of smartphone usage via anonymized Location History data to estimate actual travel times to healthcare facilities. We estimated the travel time to medical facilities starting from populated places by car, public transport, and walking. Finally, we analyzed how travel time changed during the COVID-19 pandemic. By illuminating inequities in access to healthcare and quantifying how access has changed during COVID-19, the results of this research provide a resource for policymakers tasked with optimally allocating healthcare and transportation resources under both normal and crisis conditions. More broadly, we hope his study will provide a foundation for further research and policy-based interventions focused on promoting health equity for the underserved.

## Methods

In this section, we describe methods used to develop our dataset, compare potential and revealed accessibility, assess how access to healthcare varies spatially and correlates with clinical outcomes, and estimate the impact of the COVID-19 pandemic on healthcare accessibility. Our analyses use only aggregated and anonymized data. For the purpose of this study, passenger vehicles are defined as vehicle travel excluding public transportation (buses, train, subway, etc), and the distinction in trips is made using public transportation station information (see below). The Stanford University Institutional Review Board determined that this project utilizing only non-identifiable data does not meet the definition of human subject research as defined in federal regulations 45 CFR 46.102 or 21 CFR 50.3, and waived the need for informed consent or further review.

**Medical facility database**. The inventory of geolocated hospitals and medical centers providing urgent and emergency care was extracted from Google's Maps and Search datasets in August 2019; this dataset was used to ensure consistency with anonymized trip location information, described below. Individuals can see the facilities by searching "healthcare facility" or similar on Google Maps or Search. We compared this inventory with publicly available inventories of medical facilities; coverage varies across countries (Supplementary Table 1). For potential accessibility computation, to identify the location of the nearest medical facility for all populated areas (with a population of at least 50 people based on Landscan data), we reduced gridded Landscan worldwide population density data into polygons with an average area of five square kilometers (roughly 2.23 km by 2.23 km; S2 cells level 12), computed the geographic centroid of each polygon, calculated the distance to all facilities nearby, and selected the closest one. Whereas Landscan has one of the highest population dataset coverage[21,22], exploration of other population datasets and geographical granularities may be helpful. Driving and walking potential travel time estimates could be computed for 193 regions, whereas public transportation information was not available for some regions in Google Maps and was estimated for 91 regions.

**Revealed accessibility**. To produce revealed spatial accessibility maps, we leveraged the spatial patterns of smartphone usage to estimate actual travel times to healthcare facilities across over 100 countries. Although smartphone ownership is high globally at more than 80%, we note that this distribution is not equal across several factors (see Supplementary Information, Supplementary Note 1). At a high level, this is similar to research conducted on anonymized data from internet-connected tools[23], and using passive anonymized data helps avoid issues with alternative methods such as recall bias with self-report or insufficient granularity with cell tower data[24]. Our study period spanned from January 2019 to September 2021, thereby encapsulating changing healthcare access patterns before and during the COVID-19 pandemic. All revealed accessibility metrics are aggregated by mode of transportation, and do not contain any personally identifiable information. Anonymization techniques to protect each user's activity are described in the Anonymization Strategy section below. Trips are constrained to within-country trips only.

Briefly, for each geographic region, time period, and mode of transportation, the computation of potential accessibility involves first sampling a single trip per user to remove the effect of outliers with many trips such as employees or repeated visits for multiple

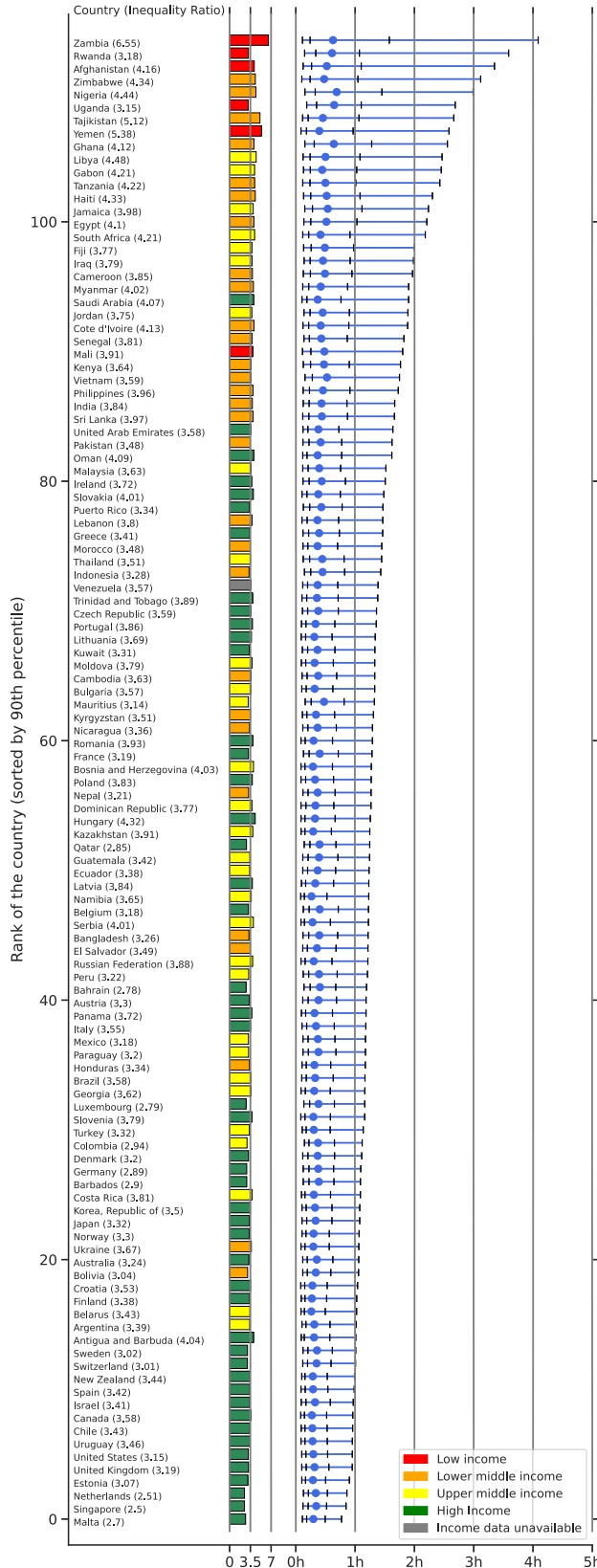

**Fig. 1 Revealed travel time to healthcare facilities using a passenger vehicle.** The inequality ratio is shown in brackets after the country name and illustrated in colored bars with colors representing income information based on World Bank classifications (see legend). Blue dots represent the median travel time in a country and vertical bars represent percentiles: 10th, 25th, 75th, and 90th. Countries are sorted by the 90th percentile. Data for other modes of transportation are presented in Supplementary Fig. 1.

summarized in Supplementary Information, Supplementary Note 2. Of note, this analysis is based on (sampling) all trips actually taken by users, and is not based on users searching for routes to a location. Next, the number of trips are summed for each of the above combinations (region, time period, transportation mode). The mode of public transportation is estimated using information such as whether segments started and ended at public transit stops (e.g., bus stops or subway stations)[26] and the breakdown is reported in Supplementary Table 2. Public transportation information was not available for some countries within Google Maps; the exact list of countries studied for each mode of transportation is listed in Fig. 1 and Supplementary Tables 3 and 4. In the following section, we provide further technical details of the anonymization techniques used to compute the 10th, 25th, 50th, 75th and 90th percentile of travel times to healthcare facilities.

**Anonymization Strategy**. Revealed accessibility is computed using aggregated, anonymized data from users who have turned on the Location History setting, which is off by default. People who have Location History turned on can choose to turn it off at any time from their Google Account, and can always delete Location History data directly from their Timeline.

The analysis starts with aggregated data across slices (geographies, time periods, travel modes). We further anonymize data across a geographic region (country), time period (calendar-year quarters, Q1 2019 - Q3 2021), and on a certain mode of transportation (passenger vehicle, public transit or walking) with $\epsilon \approx 0.9417$-differential privacy (DP)[27–31]. We do so by processing the 10th, 25th, 50th, 75th and 90th percentiles of the travel times, which are subsequently used to compute our accessibility metrics, according to the following mechanism:

1. We bound the contribution a single user can make to the distribution of travel times per geographic region, time period, and mode of transportation, to 1 by sampling a single contribution uniformly at random. This step serves two purposes. On the one hand, it limits the effect an individual can have on the aggregated metrics, and is an essential step for establishing differential privacy. On the other hand, it avoids potentially skewed estimates caused by disproportionate numbers of visits to healthcare facilities by some users. The latter consideration is intended to improve data quality.

2. We count unique users that contribute to any particular geographic region, time period, and mode of transportation (regions, periods and transportation modes without any users are excluded). We then add Laplace noise to each count using our open-sourced DP libraries[31,32]. The privacy budget is split into 6 ways to protect the 5 percentile measures enumerated above, as well as the aggregation count (the total number of visits to serve as the denominator), therefore, the resulting counts are ($\epsilon$ / 6)-DP.

3. For each geographic region, time period, and mode of transportation, whose counts after the addition of DP noise are above the threshold of k = 1000, we compute the 10th, 25th, 50th, 75th and 90th percentiles using our open-sourced DP libraries[31,33]. This threshold of k = 1000 was determined

care episodes (these trips are visible in users' Google Maps app, under the "timeline" feature). The processing of raw location data into segmented trips was explained in greater detail previously[25]. Briefly, trips are defined to start at a residence and end at a healthcare facility, and were extracted using clustering methods that enable robustness to intermediate stops. More details are

qualitatively for privacy reasons, but ensuring that a sufficient number of trips are incorporated in each estimate also serves to improve the certainty in each estimate. For example, using a normal approximation, for the 90th percentile, with 1000 trips the 95% confidence interval is small at approximately ± 2 percentiles. Each percentile aggregation is protected with $(\varepsilon / 6)$-DP. The thresholding is solely used to maintain data quality, and has no impact on DP.

Combined analysis of steps 2 and 3 via the fundamental composition theorem of DP implies $(6 * \varepsilon / 6 = \varepsilon)$-DP for the counts and percentiles[9]. Because all further analysis, including the computation of our accessibility metrics, is based on the anonymized percentiles, all user activity is protected with the specified DP parameters as a result of the post-processing properties of DP.

**Measuring changes during the COVID-19 pandemic**. To quantify changes in revealed accessibility during the COVID-19 pandemic, we analyzed the data covering each calendar quarter between January 1, 2020 and September 30, 2021. Observations from these quarters were compared to those of the corresponding (pre-pandemic) quarters of 2019 to control for normal seasonal fluctuations.

**Within-country variation in accessibility**. To examine variation within countries, we quantified inequality using percentile ratios that are similar to measures of income inequality such as the GINI index[34]. To assess the inequality of travel times within each country, we first computed travel times for the entire population, and then computed the inequality ratio, defined as the revealed travel time for the 75th percentile divided by the revealed travel time for the 25th percentile.

**Country-level outcomes data**. We correlated our measures of accessibility and inequality with published metrics reflecting health outcomes within populations, specifically national infant mortality rates and life expectancies from the World Bank[6].

**Potential accessibility**. We estimated potential travel time from every location to the nearest healthcare facility using the Google Maps Directions API, which is an automated service that returns travel directions based on input such as a requested origin, destination, and mode of transportation. Travel time was estimated separately for travel by car, public transport, and walking. Our main analysis focuses on driving estimates, as we found that passenger vehicles were the most common mode of reaching a healthcare facility urgently (see Supplementary Table 2). Travel time was estimated to represent typical traffic conditions. For each location, the population-normalized potential accessibility is computed by weighting the travel time estimates with the corresponding population density estimates from Landscan worldwide population density data[35,36]. Potential travel times were estimated in August 2019 for the analyses presented and again in July 2021 to assess robustness.

**Comparisons between potential and revealed accessibility**. Potential accessibility by car assumes car ownership, ability to borrow a car or receive a ride, or access and means to afford for-hire car services. As our results below demonstrate, potential accessibility significantly underestimates access barriers due to people needing to rely on slower or more indirect modes of transportation, such as catching a ride with multiple stops, walking, or taking public transit. We used the revealed travel data to estimate how much longer people travel *de facto* using

passenger vehicles, compared to the estimates of potential travel times by driving. We assessed the differences between the potential and revealed travel times by subtracting (aggregated) potential travel time for each country from the revealed travel time, and normalized it by the population density to produce population-level statistics for each country.

**Reporting summary**. Further information on research design is available in the Nature Portfolio Reporting Summary linked to this article.

## Results

We quantify population-weighted potential and revealed travel times to healthcare in 193 and 120 countries, respectively, based on the availability of reliable data (Methods).

**Inequalities in revealed accessibility across and within countries**. Median revealed travel time (across over 100 countries) to reach a healthcare facility using a passenger vehicle is under 44 minutes while there is considerable variation between countries (Fig. 1). Compact countries with good infrastructure, such as Malta, Singapore, and the Netherlands, generally feature the best-revealed accessibility (lowest travel times). By contrast, the longest trips to healthcare occur in countries with poorer infrastructure and/or fewer locations offering care.

To highlight disparities in healthcare accessibility, we report the 10th, 25th, 50th, 75th and 90th percentile travel times per country. For example, the median in passenger vehicle travel time in Nigeria is 41 minutes, but 10% of users travel more than 3 hours. Notably, in 17 countries, at least 10% of users traveled more than 2 hours.

We also compute the inequality ratios (defined as the revealed accessibility's 75th percentile divided by the 25th percentile; see "Within-Country Variation in Accessibility" in Methods) for each country (Fig. 1). For countries with low travel times, such as Malta, Estonia, and the United Kingdom, the inequality ratio varies between 2.5 and 3.2, indicating that the upper quartile of travel times to healthcare are 2-3 times higher than the lower quartile. Conversely, Zambia has the greatest inequality ratio of revealed travel times (6.55) in our analysis.

**Changes in revealed accessibility during the COVID-19 pandemic**. The revealed accessibility analysis captures temporal and mode-of-transport-dependent variations during the COVID-19 pandemic (first quarter of 2020 to third quarter of 2021, with the respective quarters in 2019 as the baseline). In 2020, the proportion of trips to healthcare facilities in passenger vehicles increased by 6.68% (from 70.61% to 77.28%) compared to 2019. This is accompanied by corresponding decreases in trips to healthcare facilities on public transportation (2.18%) and by walking (4.50%), see Supplementary Table 2. These trends are found in most countries, though considerable variations are observed between countries and quarters of the year.

We quantify the median percentage changes in travel time and median distance traveled, relative to a pre-pandemic baseline, across the three modes of transportation (Table 1, Figs. 2 and 3). We find that median passenger vehicle times decreased slightly in most countries, while the opposite is true for public transport and walking. Unlike travel time, the median distance traveled remains stable for walking and passenger vehicles in most countries, while it increased for public transportation.

In 14 countries (India, Mexico, Portugal, Ukraine, Belarus, Turkey, Hungary, Uruguay, Argentina, Japan, Brazil, Thailand, Russia, and South Korea), public transport travel times during the

**Table 1 Observed changes in revealed travel times observed in the second quarter of 2020 compared to the second quarter of 2019.**

| Mode of travel | Percentage Change | Absolute Change | P value |
|---|---|---|---|
| Passenger vehicle | −11.74% | Decreased by 2 minutes and 54 seconds | p < 0.001 |
| Public transport | +17.9% | Increased by 3 minutes and 53 seconds | p < 0.001 |
| Walking | +26.89% | Increased by 1 minute and 41 seconds | p < 0.001 |

Change values consist of medians and the p value was calculated using two-sided paired t-tests.

second quarter of 2020 increased by 5 minutes or more. These increases are largest in India (+13 min), Mexico (+10 min), Portugal (+9 min), Ukraine (+8 min), and Belarus (+7 min). The increases are consistent with the closure of healthcare facilities and changes in public transportation schedules.

Interestingly, trips to healthcare facilities via passenger vehicle (the most common mode) are faster relative to the pre-pandemic baseline and despite consistent distances being traveled, which suggests a decrease in traffic congestion. Conversely, given that the median distance traveled remained stable during the COVID-19 pandemic, an increase in median walking travel times suggests either a walking speed decline or an increased number of pauses due to illness -- hypotheses that need further testing. Similarly, the increase in travel times by public transportation may be attributable to a reduction in the number of operating transport lines, decreased frequency of operation, reduced capacity on public transport to maintain social distancing, and detours added to remaining lines to compensate for the reductions. Other aspects that may have impacted mode-specific travel times include reduced provider availability at the peaks of COVID-19 cases, delaying elective procedures, moving non-urgent visits to telemedicine, or closures due to staff shortages. In summary, our results suggest that populations relying on public transportation and walking to reach healthcare are more adversely impacted than populations that access healthcare services via passenger vehicle. However, additional work will be needed to better understand the causal mechanisms behind the changes we observed.

**Juxtaposition of revealed travel times and health outcomes**. We observe a strong country-level correlation between the revealed travel time to healthcare facilities and quality of health outcomes (from 2017-2018; see Fig. 4). For example, the 90th percentile of revealed passenger vehicle travel time is negatively correlated with life expectancy (Spearman's rank correlation coefficient $\rho=-0.70$, $p = 2.68\times10^{-18}$ Fig. 4a) and positively correlated with infant mortality rate ($\rho=0.67$, $p = 1.33\times10^{-16}$; Fig. 4b). Similar relationships are observed for the access inequality ratio, which is inversely correlated with life expectancy ($\rho=-0.50$; $p = 8.23\times10^{-10}$; Fig. 4c), as higher life expectancy is observed in countries with higher levels of access equality. Additionally, we find a negative correlation between the increases in travel time by public transportation during the COVID-19 pandemic and life expectancy ($\rho=-0.46$, $p = 2.71\times10^{-3}$; Fig. 4d). Multivariable analyses incorporating travel times and gross domestic product (GDP) are presented in Supplementary Table 5, with both revealed travel times and potential travel times remaining significant after adjusting for GDP.

**Inequalities in potential accessibility**. By contrast, when we then examine potential accessibility, we find the lowest potential travel times in the countries with highly developed infrastructure, where trips to healthcare facilities by passenger vehicle typically take under 30 minutes (Fig. 5). For example, in the United States, 99% of the population can potentially reach a healthcare facility by car

in less than 30 minutes. Conversely, substantially higher potential travel times to healthcare facilities by car are observed in sub-Saharan Africa. For example, in the five most populated sub-Saharan countries of Nigeria, Ethiopia, Democratic Republic of Congo, South Africa, and Tanzania, only 54%, 25%, 56%, 71% and 39% of the population, respectively, can potentially reach a healthcare facility in less than 30 minutes even when traveling by car. A direct comparison between potential and revealed accessibility (in the same countries) is described next.

**Comparisons between potential and revealed accessibility**. Figure 6 shows the difference between the potential and revealed travel times by passenger vehicle for each country for 2019. Although a significant correlation exists between potential and revealed travel times ($\rho=0.53$, $p = 1.44\times10^{-9}$), disparities were found in most countries and particularly for the longest observed trips (i.e., trips corresponding to the 90th percentile of revealed travel times). For example, among the 120 countries with revealed travel times in this study, the longest ten percent of actual trips are two, three, and five times longer than potential trips in 100, 77, and 35 countries, respectively. The population-weighted differences between potential and revealed travel times are most striking in Zambia, Rwanda, and Nigeria, suggesting that potential travel times are least accurate in lower-income countries.

**Discussion**

This study is the first to provide estimates of revealed travel times to healthcare, contrast these results with comparable estimates of potential travel times, and analyze how revealed travel time to healthcare facilities changed during the COVID-19 pandemic. Our key findings demonstrate substantial disparities in healthcare access within and across over 100 countries. Revealed travel distances offer additional insights as revealed travel times are greater than potential travel times, especially for those with the longest travel times. Compared to revealed measurements, potential travel times fail to fully capture the consequences of real-world factors such as vehicle ownership; facility closure; and utilization of more distant facilities due to specialized care, insurance considerations, or individual preference. Our findings are consistent with previous work demonstrating that spatial healthcare access estimates within a country can vary substantially depending on the estimation method[21,22,37–39].

Revealed travel times for those using passenger vehicles decreased during the pandemic, while revealed travel times for those using public transportation or walking increased. We observed an association between countries with increased travel times and countries with worse healthcare outcomes, including higher infant mortality rates and lower life expectancy. Overall, our study confirmed the utility of aggregated and anonymized user-generated data to further our understanding of key global health issues.

Prior research has enabled the mapping of healthcare facilities and computing potential travel times at fine spatial resolution globally[19,40]. Most previous work on geographic accessibility

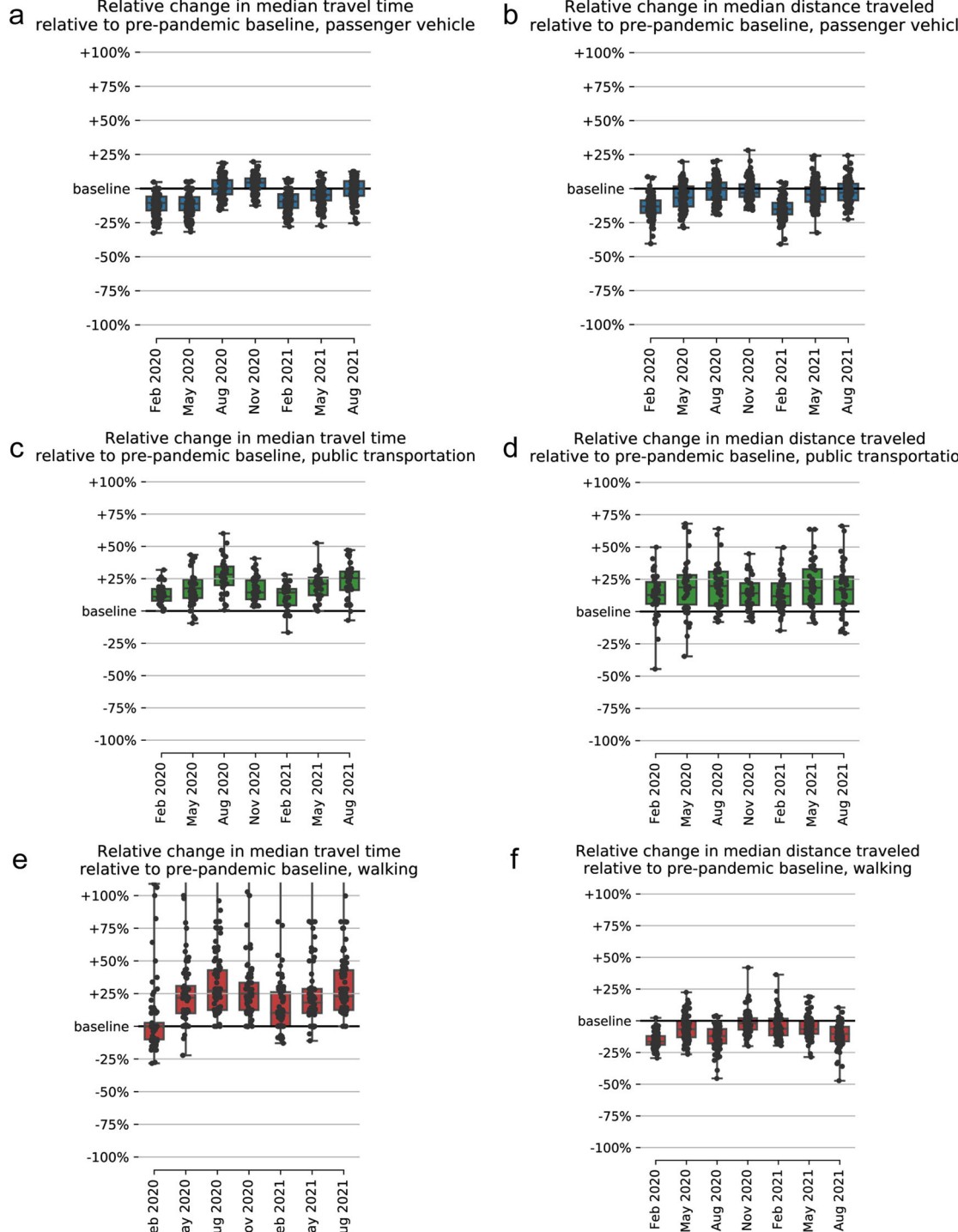

**Fig. 2 Relative change in median travel time and median distance traveled compared to the pre-pandemic baseline (corresponding quarter in 2019).** The three rows represent data for passenger vehicles (**a, b**), public transportation (**c, d**), and walking (**e, f**), respectively. The panels on the left (**a, c, e**) depict change in travel time, and those on the right (**b, d, f**) depict change in distance traveled. Box-plots summarize the revealed time statistics across countries. Boxplots edges represent the 50th (center line), 25th, and 75th percentile (box limits), whereas the whiskers extend to the minimum and maximum values but no further than 1.5 times the interquartile range.

employed measures of potential access, such as straight-line distance, the volume of services provided relative to the population's size, the proximity of services provided relative to the population's location, the estimated travel times, or a combination of those factors. Notable examples include the Enhanced Two-step Floating Catchment Area (E2SFCA) method and the Index of

Spatial Accessibility (ISA)[19,40]. However, the application of existing methods has been limited to specific countries and specific healthcare services[5,41–43], low-resource settings[44,45], or only capture certain dimensions of potential accessibility worldwide[4]. Similarly, geospatial studies aiming to measure healthcare access during COVID-19 pandemic have largely been limited to specific

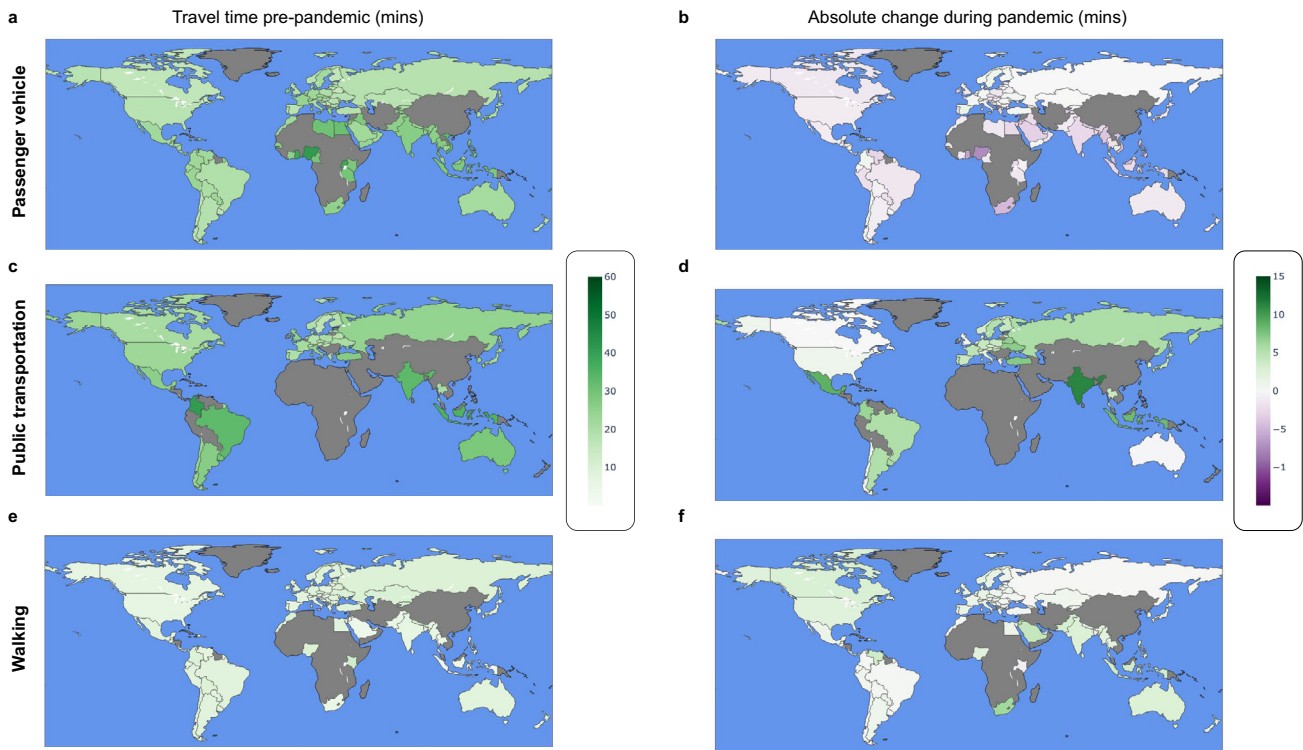

**Fig. 3 Pre-pandemic median revealed travel times to healthcare and absolute changes therein during the COVID-19 pandemic.** Panels **a, c** and **e**, depict median revealed travel times pre-pandemic (2019), whereas panels **b, d,** and **f** depict absolute changes during the COVID-19 pandemic (2020–2021, see Methods). Gray color indicates countries or regions not included in the study; see Fig. 1 and Supplementary Table 3 for the list of studied countries.

countries and relying on potential accessibility-based catchment-area methods[46]. The current paper is an advance on this literature as it explores both potential and revealed accessibility. In particular, potential accessibility fundamentally does not account for the choice of transportation modality, the choice of healthcare facility, or the day/time of travel, whereas revealed accessibility incorporates aggregated and anonymized real-world travels. Finally, we show that revealed accessibility correlates with important health outcomes across countries, even after adjusting for potential accessibility and wealth (GDP), indicating that accessibility is an important independent factor to track, understand, and improve.

The COVID-19 pandemic caused a global crisis and created unprecedented disruptions in various spheres of people's lives[47–49]. While travel time to healthcare is just one aspect of multifaceted healthcare delivery systems, our analysis is unique in providing a directly measured assessment of pandemic-related disruptions in over 100 countries. We found substantial variation in travel times during the COVID-19 crisis, compared to corresponding pre-pandemic conditions. The increases in travel time during the pandemic are likely to disproportionately impact more vulnerable populations, in particular those with low car ownership rates or ready access to car travel. In this sense, our analyses showed that inequalities in access may have been exacerbated during the COVID-19 crisis.

These results have implications for public health policies. Individuals are less likely to seek healthcare when long journeys are required to obtain the services[50]. By enumerating disparities between countries, this research highlights areas that could benefit most from improving access to healthcare, which is achievable by adding more public transportation lines, subsidizing transportation, adding healthcare services closer to where populations live, and introducing mobile clinics. This study also quantifies, for the first time, the travel times to healthcare facilities for a number

of transportation modes. This information better contextualizes the costs of travel and thereby strengthens the evidence available to policymakers responsible for directing healthcare infrastructure investment. Our results echo previously identified knowledge gaps and calls for more fine-grained and temporally-aware accessibility metrics, more sophisticated geocomputational tools to operationalize such metrics, and improved measurement of inequalities[51].

Global-scale and spatially representative datasets that quantify metrics relevant to healthcare accessibility are relatively scarce, despite being necessary for quantifying disease burden and healthcare demand. The results of this research help bridge this gap by introducing a set of metrics based on empirical data. As such, this research and subsequent analyses that build upon our methodology have the potential to inform public health makers tasked with formulating policy and directing resources towards areas or public health needs most in need. Future work is needed to apply these methods at regional and metropolitan level, in order to make the resulting insights actionable to local stakeholders. For example, health planners could consider revealed travel times in planning for healthcare service provision, optimizing public transit routes, or spatially targeting health promotion campaigns. Finally, as demonstrated with the COVID-19 analysis, the dynamic nature of revealed travel times allows them to serve as sentinel datasets for monitoring changing patterns of spatial access to healthcare.

This study has a few limitations. First, our inventory of healthcare facilities (based on Google Maps) may be incomplete or inaccurate, with the quality and completeness of the dataset likely to vary between countries (Supplementary Table 1). Second, to preserve user privacy, we used a radius of 500 m around each facility, which potentially included some non-healthcare-seeking trips in the analysis (the analysis may also include trips by people who visit hospital patients but do not receive healthcare themselves). This data may skew the

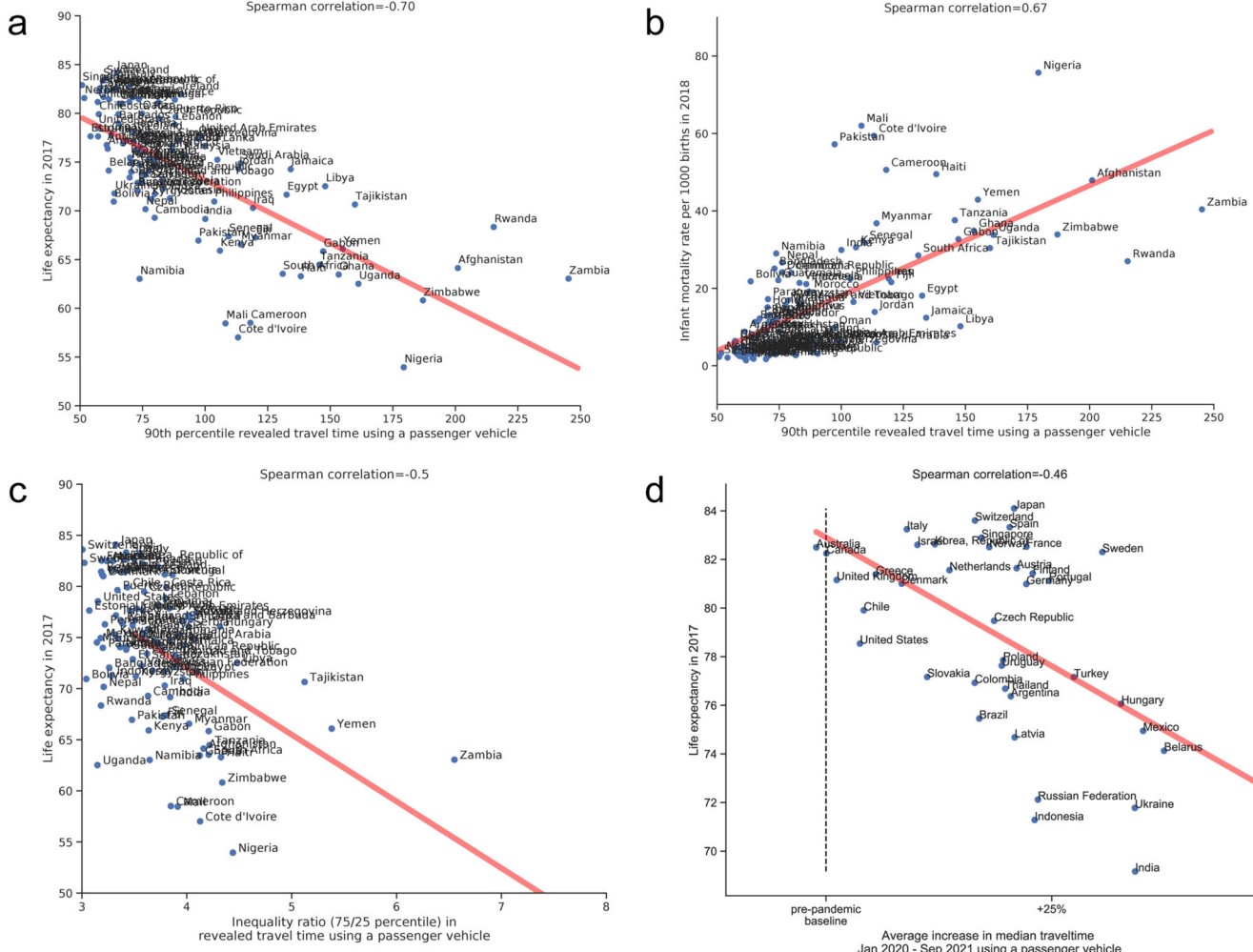

**Fig. 4 Spearman's rank correlation of revealed accessibility metrics with health outcomes. a** The correlation between the 90th percentile of the revealed travel time in a passenger vehicle, with life expectancy in 2017. **b** The correlation between the median revealed travel time and infant mortality rate in 2018. **c** The correlation between the inequality ratio and life expectancy in 2017. **d** The correlation between average increase in public transport travel time during the COVID-19 pandemic, and life expectancy in 2017. Note that the red line is the linear line of best fit shown for illustrative purposes; the Spearman correlation is based on the rank and is not linear with respect to these axes. Quantitative analyses of health outcomes and revealed accessibility while controlling for potential accessibility and wealth is presented in Supplementary Table 5.

distribution of modes of transportation people use to get care if, for example, someone in need of care travels faster than normal out of urgency or more slowly if encumbered by illness or injury. Third, we excluded from the analysis countries with a low number of trips to healthcare facilities, so additional countries will need to be covered in future work as smartphone ownership and the coverage of healthcare facilities increases. Fourth, this analysis was a general examination of healthcare access across many facilities, and because information on facility-level service offerings were not available, this analysis does not provide granular insights about travel times to specific services such as primary care or specialty services. Furthermore, patients may choose to receive healthcare not in the nearest facility, if a different facility is covered by insurance or was recommended by the referring physician or a family member, if the nearest facility is closed at that time, or if care is delivered in the home or in a community setting. Similarly, exogenous factors such as the COVID-19 pandemic also influenced the type and mode of care (e.g., preventative vs. urgent, in-person visit vs. telemedicine, lockdowns and ability to travel to the care facility), and this anonymized analysis measures the net effect on in-person travel without the ability to break down the data based on these factors. And fifth, travel time is only one of the numerous barriers to

healthcare access. For example, lack of health insurance, financial constraints, or stigma could prevent people from receiving care.

Another potential limitation of this research is the representativeness of the input data used to estimate revealed travel times, specifically whether those who own and use GPS-equipped mobile phones and opt into location sharing[24] are a representative sample of the general population. Evidence suggests that, particularly in developing countries, phone ownership skews towards wealthier individuals, younger ages, and males[52]. In turn, if phone ownership is more common in urban environments where travel times are shorter, the inequities observed in this study may constitute underestimates of the true state. As such, the results of this work should be interpreted cautiously in cases where the movement patterns of a population of interest may differ from those in our study population. Similarly, specific analyses on the subnational level such as urban vs. rural differences or local regions or time spans of interest will need to be explored in future work and in collaboration with local public offices. We further discuss this issue and data from the literature in the Supplementary Information, Supplementary Note 1. Lastly, although we demonstrated the country-level association between

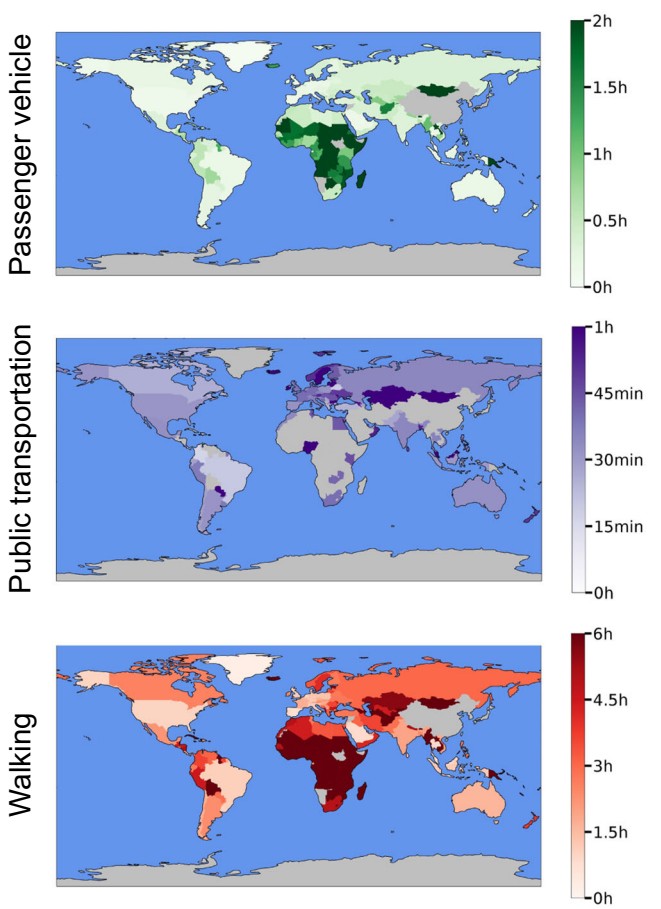

**Fig. 5 Estimated potential travel time for different modes of transportation.** Gray indicates countries for which data were not available due to privacy and data availability/quality (Methods). Comparisons between potential and revealed travel time are presented in Supplementary Fig. 2a.

the level of healthcare access and quality of outcomes, it was beyond the scope of this analysis to fully address the causal link. For example, the changes in mode-specific travel times during 2020 were likely multicausal, spatially heterogeneous, and stemmed from a combination of government policy and individual decisions. Similarly, individuals' socioeconomic circumstances before versus during the pandemic and choices regarding the mode of transport may have influenced the trends observed. Future research is needed to better characterize the factors affecting travel times to healthcare, and such analyses will offer a great resource for considering healthcare access equity when responding to current and future public health crises.

## Conclusion

Spatial accessibility is an essential dimension of healthcare systems, and our analysis shows substantial spatial and temporal disparities in this metric within and across countries. We demonstrate the utility of user-generated mobility data to produce privacy-preserving estimates of revealed travel times to healthcare facilities at global scale. These results capture substantial changes in healthcare accessibility as the COVID-19 pandemic unfolded, and indicate that these changes were not distributed equally across populations within countries. Our

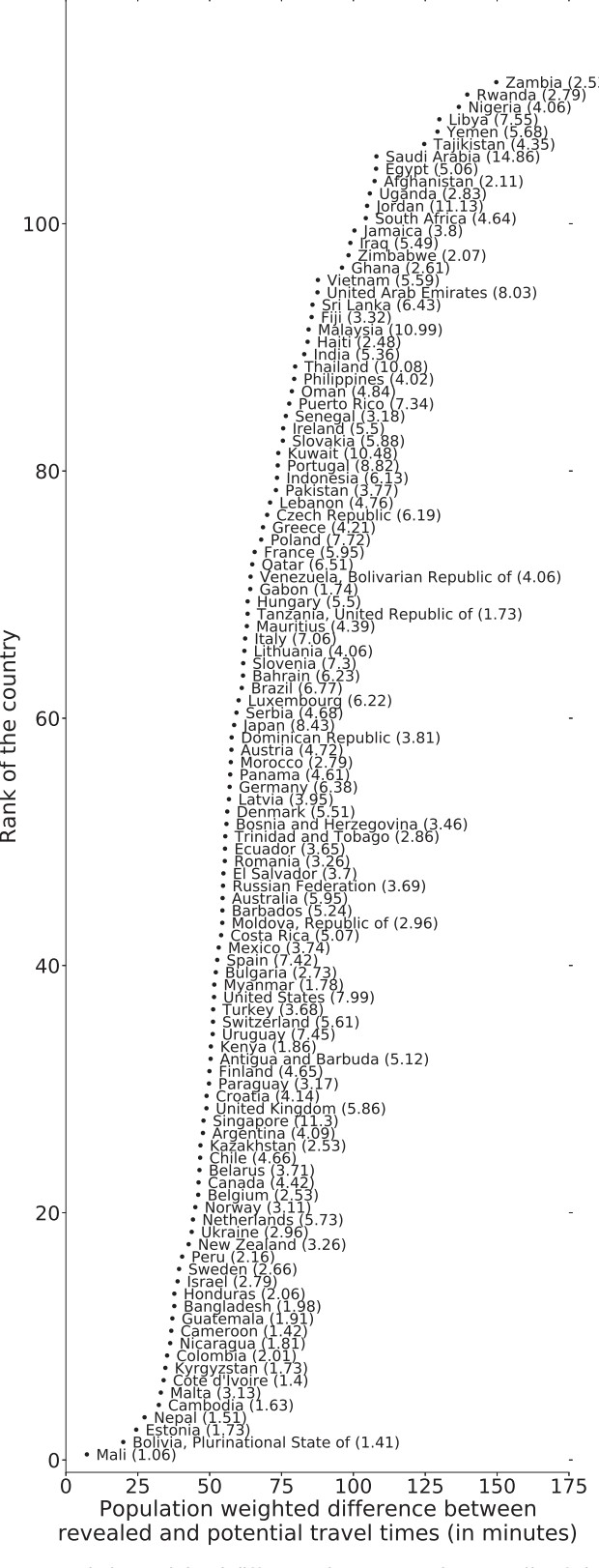

**Fig. 6 Population-weighted difference between 90th percentile of the revealed travel time and potential travel times (in minutes) for the time period of Jan 2019–Dec 2019.** The factors by which revealed travel times differ from potential are shown in parentheses after the country name. Corresponding analysis for revealed and potential travel distances is presented in Supplementary Fig. 2b.

insights are an important step towards enabling a greater understanding of barriers to care for the global population.

## Data availability

The following aggregated and anonymized data reported throughout this manuscript is made available as part of Supplementary Data 1: revealed accessibility i.e. distribution of times (by quantiles) to reach the nearest medical facility, by country and mode of transportation (as reported in Fig. 1); COVID-19 analysis (by quarter over 2020-2021) i.e. percentage change in median travel time and median distance traveled, compared to the pre-pandemic baseline, reported globally (Fig. 2) and pre-pandemic median travel time and absolute change therein during the pandemic, by country and mode of transportation (Fig. 3); country-specific inequality measures, and correlation coefficients between the measures of revealed accessibility and healthcare outcomes (Fig. 4); potential accessibility i.e. expected time in minutes to get to the nearest medical facility, by country and mode of transportation (Fig. 5); and population-weighted difference between the revealed and potential travel time (in minutes), per country (Fig. 6).

## Code availability

Code that produces reported results based on aggregated and anonymized data is provided in Supplementary Data 1.

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

## Acknowledgements

We thank Amit Chibber, Katherine Chou, Iz Conroy, John Davis, Alex Fabrikant, Bryant Gipson, Matt Hancher, John Hernandez, Vivien Hoang, Michael Howell, Mansi Kansal, Dennis Kraft, Lisa Lehmann, Allie Lieber, Rafael Lozano, Kori Meehan, Leeron Morad, Christopher Murray, Andrew Oplinger, Genevieve Park, Alvin Rajkomar, Thomas Roessler, Troy Sauro, Eric Tholome, Andrew Tomkins, Doris Wang, Janet Whiteman, Gregory Wellenius, Brian Williamson, Royce Wilson, Lauren Winer, Greg Wolff, Marissa Urban, and Ashley Zlatinov for their help and advice.

## Author contributions

K.G., C.K., D.W., S.B., T.S., K.S., and E.G. contributed to the development of the concept. The authors affiliated with Google had access to the aggregated and anonymized data. K.G., C.K., and S.B. computed the data. All authors (K.G., C.K., D.W., S.B., Y.L., T.S., K.S., and E.G.) contributed to the design of the experiments. K.G., C.K., and S.B. analyzed the data and performed statistical analysis. All authors (K.G., C.K., D.W., S.B., Y.L., T.S., K.S., and E.G.) contributed to writing the manuscript. KG, CK, and DW contributed equally to this manuscript and therefore are listed as co-first authors. K.S. and E.G. jointly supervised the work.

## Competing interests

The Authors declare no Competing Non-Financial Interests but the following Competing Financial Interests. KG declares no Competing Financial Interests. DW is funded by the Bill and Melinda Gates Foundation. The funders had no role in study design, data collection and analysis, decision to publish, or preparation of the manuscript. CK, SB, YL, TS, and EG are current or past employees of Google and own Alphabet stock. KS declares no Competing Financial Interests.
