## [Peer Review File · Communications Medicine]

Reviewers' comments:

Reviewer #1 (Remarks to the Author):

The authors describe a novel method for mapping geographic accessibility using google data. In addition, they compare two generated travel times called the revealed and the potential. The paper is quite interesting and I believe contributes to the gaps that exist in literature.

I have several points for the authors to respond to or consider including.

1. I find the paper to be quite long, with very many messages captured and difficult to maintain an understanding of what the key message is. First, there is a description of accessibility in a novel way which I think is quite good and has a lot of information. Second, there is a comparison between revealed and potential accessibility which is also quite novel and useful. Third, there is comparison of accessibility over time to unpack whether there were any differences before and during COVID. Finally, there is a comparison between accessibility and outcomes such as mortality. I think the authors should consider prioritizing a specific theme and focus more on say comparison between the two metrics and using the choice of one to compare either with outcome or get the impact of COVID 19.

2. Conceptually, government restrictions to movement during the COVID potentially prevented patients from using facilities. I therefore think that it would be useful to not attribute this to reducing access to facilities (i.e access remained the same in where facilities were not increased in number) but rather less people were likely to use facilities. Thus, I would suggest rephrasing the conclusion accordingly. For example the conclusion states the following;

“These results capture significant changes in healthcare accessibility as the COVID-19 pandemic unfolded, and indicate that the impacts to healthcare access were not distributed equally across populations within countries”

But should show that restrictions acted as a barrier to utilization of facilities, potentially even in areas with access

To a statement that captures the fact that conceptually, during covid government imposed travel restrictions reduced made people less likely to use facilities and this inturn lead to less people being captured in the realised travel.

However, few systematic efforts have characterized healthcare accessibility at the global scale and none have empirically assessed how this key public health metric has been impacted by the pandemic.

Methods

The authors state the below;

“To identify the location of the nearest medical facility for all populated areas, we reduced gridded Landscan worldwide population density data into polygons with an average area of five square kilometers, computed the geographic centroid of each polygon, calculated the distance to all facilities nearby, and selected the closest one.”

1. What are the chances that there are multiple facilities within a 5 square km grid and this would definitely affect the accessibility metric

2. In most low-income countries, access to smartphones is poor and therefore using such a method proposed would mean the results are skewed. I would suggest putting it across that this work is only beneficial in settings where access to smartphones is high and also they have location services enabled

“Briefly, for each geographic region, time period, and mode of transportation, the computation of potential accessibility involves first sampling a single trip per user to remove the effect of outliers with many trips (these trips are visible in users’ Google Maps app, under the “timeline” feature).”

3. Sentence is not very clear and why the sampling was done. Does it mean a single user had many trips from residence to facility and therefore one was taken? Need to clarify the statement.

4. Methods needs abit more explanation rather than packing into supplementary file. For example no information is provided on how it was possible to distinguish the use of public transport from personal vehicles and this would have an impact on accessibility estimates

5. I think the Fig A3 on revealed travel time in the supplementary file should be in the main manuscript. Also Would suggest grouping by smartphone usage/availability in countries for easier comparison and understanding of the error bars.

Reviewer #2 (Remarks to the Author):

Does the manuscript have technical or conceptual flaws that should prohibit its publication? If so, please provide details.

No, but I feel this research could be better located within the existing literature, particularly more recent studies on measuring accessibility to health care (in general or specific services) and the impact of the pandemic in access/availability of health services (very little literature on this is cited or explored). I would also suggest that justification of the choice of datasets is needed, and explanation of which countries are considered in the analysis (particularly as there are many references to the analysis being global in nature).

Are the conclusions original? If not, please provide relevant references.

The analysis provides new insights into potential and “realised” access to health services, considering both a pre-pandemic and during pandemic time period, across many countries. I would question whether the paper’s claims of being useful for understanding where to locate new facilities and services, are really the case, as the results presented are all at national level – if the analysis provides sub-national results, it would be interesting to include these also.

Do you feel that the results presented are of immediate relevance for many people in your own field or for a broader audience?

I think the results will be of interest to a wide audience, however the presentation of results being limited to the national level will likely limit its interest for those actually involved in planning health resource allocation at a more local level. The limited explanation of which countries are included in the analysis also makes it difficult to interpret how widely the results are applicable.

If applicable, does the manuscript reporting follow ICMJE and EQUATOR recommendations? If not,

please provide details.

Not applicable

If you recommend publication, please outline briefly what you consider to be the outstanding features.

If the issues outlined in these comments are addressed satisfactorily, I would recommend publication – most relate to the justification of datasets used, the time periods/countries considered and the situating this research better within the existing literature. This paper's main strength is the use of Google datasets that are not widely available to researchers, and the new insights that this presents, across multiple countries.

If you feel that specific additional experiments would strengthen the case for publication in Communications Medicine, please provide suggestions.

As described above, inclusion of analysis for sub-national locations would strengthen this paper.

General comments

Thank you for the opportunity to review this paper. It is an interesting analysis utilising Google location history data across many countries, and Google routing applications to calculate potential travel times. This is explored in two time periods: just before and during the COVID-19 pandemic. The following are general comments which I hope can help make the analysis and interpretation of results clearer to readers.

- Please clarify if accessibility is being measured to all healthcare facilities (defined how?) or only a subset offering. In the fourth paragraph of the introduction, you state "the inventory of geolocated hospitals and medical centers providing urgent and emergency care..." which suggests it's just a subset of health facilities being considered, but throughout the rest of the paper, you make reference to healthcare, healthcare facility and medical facility. Then in the limitations, you write "because not all facilities offer all services, cannot be used to assess travel times to specific services", which seems to contradict your initial statement. It would help those reading the paper if you make this clear early on in the paper and then use consistent terminology throughout.
- At the end of the abstract you state that the results can be used by policy makers for directing resources and formulating policies – suggest indicating that these results should be considered in conjunction with other sources, i.e. not used in isolation.
- The geographic scope of the paper should be clarified. There is mention of this being a global analysis, but then at the start of the results then you state "we quantify population-weighted potential and revealed travel times to healthcare in 193 and 120 countries respectively", without explanation of why and which countries have been included in this analysis. I presume this is probably due to data coverage, but this should be clearly stated. Could the inclusion of only a subset of countries potentially alter results compared to a truly global analysis?
- Did you do any analysis to understand the coverage of the health facility data used in comparison to other sources e.g. OSM or WHO health facility list (Africa only)?
- A few comments related to figures:
 - o Figure 3 – the world maps are too small to see any changes for small countries. Please also indicate what grey is (no data?). Also consider a colour scale where 0 is not the same colour as the background/non-landmass
 - o Figure 3 caption – consider stating the time period for "pre-pandemic" and "COVID-19 pandemic"
 - o Figure 4 – the crowded labelling of data points in a, b and c makes this very difficult to read or interpret. Please revisit
- In the limitations section you ask whether users are a representative sample of the general population – what does the literature indicate about this?

- In the final paragraph of the introduction you state “we leveraged the spatial patterns of smartphone usage” – this is quite a broad statement, and would suggest a data source that represents a wider population than is captured by Google Location History. Please make this clearer.
- In the first paragraph of the results, please clarify the mode of transport for the under 44 minutes result.
- In the results section, when stating results for “during the pandemic”, please be clear the time period being referred to.
- In the third paragraph on page 4, when you discuss changes in travel time, have you considered whether socio-economic impacts may have changed the make-up of e.g. who was walking?
- In the final paragraph of the results (page 5), there’s mention of biggest differences in sub-Saharan Africa, which I presume is referring to the top 3 countries in figure 6 being Zambia, Rwanda and Nigeria. If this is case, consider referring to these countries by name. There are other sub-Saharan African countries ranked much lower in figure 6 (e.g. Kenya, Mali, Gabon). From a quick count there appears to be only 11 SSA countries shown in figure 6, whereas there are over 50 countries in SSA, so difficult to know if similar patterns would be seen in other countries not included in the analysis.
- In the analysis comparing travel times and health outcomes (final paragraph page 4), it’s not immediately clear at what geographic scale this comparison is being made (only at national-level I think?). And the health outcome indicators are presumably only from pre-pandemic period? I think it’s important to explain this more explicitly.
- The Methods section is currently right at the end of the paper (after figures and references). My understanding is that Comms Med papers have the methods section integrated into the main text, so I presume this will be changed? If not, then I would suggest adding some details earlier in the paper explaining how the pre-pandemic baseline was defined and the dates it covers, similarly for the period of the COVID-19 pandemic considered. Similarly I would suggest adding an explanation of the source of population numbers and populated places earlier in the text to help readers have a clearer understanding of the analysis.
- When comparing results from during the pandemic to pre-pandemic baseline, was there any consideration around differences in lifting of restrictions in different countries?
- In the Methods section, the description of the processing of the LandScan data needs to be revisited and the explanation improved. When you state “we reduced gridded LandScan worldwide population density data into polygons with an average area of 5 sq. km”, I presume you are meaning that the population density raster was resampled (i.e. the grid cell size changed) and the output grid cell size was increased? Please clarify whether the resampled grid cell size was 5x5km (i.e. 25 sq. km) or actually ~2.23x2.23km (i.e. 5 sq.km) – currently you are indicating the latter, but I think you may be meaning the former?
- Is the above the process for defining “populated places”, if so please state this or if not please explain how populated places are defined.
- Please justify the choice of LandScan population dataset? You may want to consider recent work by Hierink et al. (2022) which explores the impact of utilising different gridded population datasets on accessibility measures.
- Much of the literature cited is relatively old (15/31 references are from 2011 or earlier). Please try to ground this work in more recent literature as there has been considerable work in the last decade on measuring accessibility to health facilities/different types of healthcare. Just a few of the many recent papers/chapters to consider:
 - o Ouma et al. (2021) Methods of Measuring Spatial Accessibility to Health Care in Uganda, https://doi.org/10.1007/978-3-030-63471-1_6
 - o Hierink et al. (2022) Differences between gridded population data impact measures of geographic access to healthcare in sub-Saharan Africa, <https://doi.org/10.1038/s43856-022-00179-4>
 - o Bouanchaud et al. (2022) Comparing modelled with self-reported travel time and the used versus the nearest facility: modelling geographic accessibility to family planning outlets in Kenya, doi:10.1136/bmjgh-2021-008366

- o Dotse-Gborgbortsi et al. (2020) The influence of distance and quality on utilisation of birthing services at health facilities in Eastern Region, Ghana, doi:10.1136/bmjgh-2019-002020
- o Wigley et al. (2020) Measuring the availability and geographical accessibility of maternal health services across sub-Saharan Africa, <https://doi.org/10.1186/s12916-020-01707-6>

Reviewer #3 (Remarks to the Author):

General Comments.

The manuscript presents a novel analysis of the impact of COVID on geographic accessibility, comparing hypothetical estimates with actual estimates using Google data. However, the validity of using smartphone data to track movement for global comparisons is a concern, due to unequal distribution of smartphones, which limits the representativeness of the results. The authors acknowledge this limitation but further clarification is needed to justify the use of global comparisons. Additionally, the exclusion of information from several countries raises questions about the accuracy of the global analysis.

Reviewer comments:

1. Introduction:

- The focus of the work should be revised to reflect that it is not solely on comparison between pre- and post-pandemic comparison.
- The introduction needs to be more focused on Geographic access and the comparison between modelled and actual travel times/distance.
- The introduction speaks a lot about the broader concept of health access in general and not enough background to the actual work presented.
- Include more information on the importance of comparing potential vs revealed access or the discussion underpinning the hypothesis around why COVID would affect access needs to be explained.

2. Focus of the work:

- It is unclear what the focus of the work is (comparison between potential and revealed or how geographic access has changed over time in relation to COVID).

3. Changes in transport modes:

- The fact that people may have stopped using public transport in favor of private vehicles to reduce exposure or the fact that public transport was stopped in some countries needs to be accounted for.
- COVID restrictions varied across countries and over time which may influence broad statements around

4. Results:

- The inclusion of other factors like telemedicine and delayed elective procedures and their impact on mode-specific travel time needs to be explained. Presumably these don't contribute to the travel time metric because they are non-occurrences
- The value of including the comparison of revealed travel time to health outcomes is unclear, especially because of the estimates of health outcomes in from 2017 and especially concerning the comparison in figure 4d.
- Also, I am not sure this piece of analysis adds to the work presented

5. Distinguishing between a passenger vehicle and public transport:

- Define what is meant by passenger vehicle vs public transport. Public transport also include passenger vehicle
- The method used to distinguish between a passenger vehicle and public transport, especially in

countries where passenger vehicles are similar to public transport, needs to be made clear.

6. Health facility database:

- The databases used for the health facility database need to be specified and completeness will vary geographically and between developed and developing countries. This needs to be acknowledged.
- The data extraction method (from Google maps and search datasets) needs to be explained. There are several available health facility databases available why restrict this to a single comparison.
- Indicate the temporal timepoint reference for health facility databases used

7. Accessibility modeling:

- More detail needs to be provided on how the accessibility modeling was done.
- Was the computation of revealed accessibility restricted to country borders?
- The countries from which the analysis was able to be run needs to be indicated in the main text methods and why some countries only had information on PV and not for public transport
- Clarify what Google Maps Directions API is for non-familiar users

8. Figure 3:

- The purpose of the grey areas in Figure 3 needs to be explained.
- The countries for which the authors had data and the consistency of this data across different modes of travel needs to be mentioned in the methods.

9. Figure 5:

- The county estimates shown in Figure 5 need to be specified.
- The missing data on public transport needs to be explained.

10. Use of smartphone data:

- Concerns around the use of smartphone data for revealed travel time (as only 46% of populations in SSA have a smartphone and an even smaller percentage have location enabled) need to be addressed.
- The potential for non-representative populations and problematic cross-country/cross-regional comparisons need to be expanded on in the limitations. It potentially compromises the comparisons presented in the paper
- Hence wonder if it makes more sense to present this data by regions

Reviewer #4 (Remarks to the Author):

I congratulate the authors for the impressive and innovative work that they present. The paper is novel and presents the use of new technologies in the field of geographical accessibility modelling. The comparison between potential and revealed travel time adds new realism to accessibility modelling and reveals important differences. The effect of COVID-19 on revealed travel times to health facilities shines light on the impact of large-scale health emergencies on health seeking behavior and challenges in accessing care. I have a few concerns regarding the paper that I would like to share in the points listed below and I encourage the authors to address these to further improve the manuscript. I remain at the authors disposal in case they have any questions considering the below points.

1. The authors state that countries with a low number of trips to health facilities were excluded. However, the authors do not state what number was considered as the threshold and how many users were analyzed/included per country. The authors should state what number of users was considered representable and reliable and why. The number of users may also vary a lot between countries and therefore largely impact results.

2. The authors state a high inequality in revealed accessibility in several low- and middle-income

countries (e.g. Zambia, Rwanda, Afghanistan, Zimbabwe and Nigeria). I think that much of the variation found in countries in sub-Saharan Africa and Central Asia is impacted by a higher number of users in urban areas and a low number of users in remote areas, potentially causing outliers to have a greater effect. This will likely cause even larger variation between the percentiles than can be already expected. Did the authors consider making any adjustments or corrections for the number of users in the different regions of a country? Or distinguishing urban/rural differences?

3. All of the results presented in the main manuscript and the supplementary material are on a national scale. The aggregation to national level complicates the identification of sub-national variation and thus gaps in service delivery which are extremely important. I encourage the authors to present some of the results sub-nationally or to elaborate why results could only be presented at national scales.

4. Another important aspect is the change in users during the pandemic. Considering that many of the high-income countries and even some low-and middle-income countries had lockdowns in place, preventing people to go outside and thus having to use Google Maps or other related services. Was there a change in users and if so, what is the impact of the change in the number of users? How do the authors ensure that this doesn't impact their results?

5. Many hospitals during the lockdown remained closed for visitors. How did the authors make sure to factor-in a potential difference in the noise of Google users going to other places falling within the 500m radius?

6. The authors present the impact of revealed travel times on health outcomes during the lockdown. Yet, the manuscript does not describe where this health outcome data comes from, nor how it was analyzed. The authors should describe in more detail how this analysis was done and with what data if they decide to present these results. These health outcomes might not only be attributable to travel times, but also to quality of care, availability of staff, etc. during the pandemic. This should be mentioned with more emphasis.

7. The health facilities included in the analysis were fetched using Google maps and a comparison was made with healthsites.io. Even though these platforms provide useful data in the absence of government data or other reliable data sources, there are often large discrepancies between the different facilities lists (South et al., 2021. <https://wellcomeopenresearch.org/articles/5-157/v1>), implying that some important facilities may have been left out in the analysis. Especially in countries where Google maps does not provide a comprehensive overview of all services, like LMICs. Did the authors consider making a comparison between their facility list and master facility lists for some countries that are publicly available? If so, what were the results? I would highly recommend that especially for countries in LMICs the authors do some cross-validation checks with governmental facility data.

8. The authors decide to have 193 countries with potential travel time and 120 countries with revealed travel time. I would highly recommend making the comparison only between countries for which you have both revealed and potential travel time and thus reducing the total number to 120. I would expect that for many countries with potential travel time that do not have revealed travel time, the actual travel time may be much larger in rural areas, where at the moment more traditional friction layers may be more representative of reality than the Google API is now known to be.

9. In the discussion, the authors state: "potential travel times fail to fully capture real-world conditions reflecting vehicle ownership; facility closure; and utilization of more distant facilities due to specialized care, insurance considerations, or individual preference". Yet, revealed travel time also fails to capture

these factors. A Google user might be a taxi driver. And if revealed travel time manages to capture facility closure, how did the authors identify closed facilities? Did they assume facilities were closed when they could see the user by-passed a more near facility? This currently remains unclear in the text and should be addressed.

10. The authors used aggregated measures of LandScan population data at 5 km². I'm wondering why the authors decided to go for LandScan when more refined estimates are available from other data sources and differences between population datasets are found to be large (Hierink et al., 2022). WorldPop refined the population estimates in their constrained layers by using building footprint data. By using building footprint data, the population estimates are constrained to places where actual settlements are detected, minimizing the possibility of having population counts in extremely uninhabitable and unpopulated places. In addition, why did the authors decide to aggregate this data and not calculate the nearest facility for each population point/grid cell and then take the median or average of all these distances and travel times? I encourage the authors to elaborate on their decision-making behind this and highly advise them to consider another data source and analysis method for the population aggregation.

11. In the manuscript the authors state: "Conversely, substantially higher potential travel times to healthcare facilities by car were observed in sub-Saharan Africa." Yet only a very small number of sub-Saharan countries were included in the study. At the moment the study lacks a strong foundation on how the authors made sure that the data they had for LMIC countries was representable and why in this case they did not consider transport by foot. The authors do explain that smartphone and car ownership is likely to be biased towards individuals with higher incomes and livelihoods. This is probably not representable of the most vulnerable. The authors need to strengthen this aspect throughout the manuscript. Including as well that the impact of COVID-19 and the measures taken were very different between countries. This aspect is not emphasized at the moment.

12. The legend of Figure 3 should also included a symbol for the grey countries.

Response to referees

Dear reviewers,

We are pleased to submit a revision of our manuscript "*Revealed versus Potential Spatial Accessibility of Healthcare and Changing Patterns during the COVID-19 pandemic*" for publication in Communications Medicine.

We are grateful for detailed and constructive feedback which helped refine and improve the manuscript. For full details, please refer to the point-by-point response attached.

In particular, we paid attention to revising the manuscript to address issues related to contextualizing the paper within recent spatial access literature (R2.1, R2.23), improving clarity on the objectives of the paper (R1.1, R3.6), clarifying geographic scope of the work (R2.8, R3.1, R4.1, R4.8), discussing the uneven distribution of smartphones (R1.4, R2.11, R3.1, R3.16, R4.11), clarifying the choice of gridded population datasets and subsequent processing (R2.20, R2.22, R1.3), describing the sources of health facilities data (R3.12), and providing additional details on the computation of travel time (R1.6, R3.13). Please see attached the complete reviews and our response describing how we addressed each individual comment.

Yours sincerely,
Evgeniy Gabrilovich, on behalf of the authors

Reviewers' comments:

Reviewer #1 (Remarks to the Author):

R1.1: The authors describe a novel method for mapping geographic accessibility using google data. In addition, they compare two generated travel times called the revealed and the potential. The paper is quite interesting and I believe contributes to the gaps that exist in literature.

I have several points for the authors to respond to or consider including.

1. I find the paper to be quite long, with very many messages captured and difficult to maintain an understanding of what the key message is. First, there is a description of accessibility in a novel way which I think is quite good and has a lot of information. Second, there is a comparison between revealed and potential accessibility which is also quite novel and useful.

Third, there is comparison of accessibility over time to unpack whether there were any differences before and during COVID. Finally, there is a comparison between accessibility and outcomes such as mortality. I think the authors should consider prioritizing a specific theme and focus more on say comparison between the two metrics and using the choice of one to compare either with outcome or get the impact of COVID 19.

Response: We thank the reviewer for the positive comments. We indeed captured a lot of information in this manuscript. As noted by other reviewers, the title is not reflective of this and we have updated to “Revealed versus potential spatial accessibility of healthcare and changing patterns during the COVID-19 pandemic”.

We have also edited the introduction to add more context on the link between these data: “Whereas potential (theoretical) accessibility based on routes and population density is helpful in planning, it is important to understand if reality (revealed accessibility) deviates from theory, and by how much. Understanding this difference and how it is influenced by crises (such as the COVID-19 pandemic) can help further refine planning and catalyze efforts to improve healthcare accessibility.”

R1.2: 2. Conceptually, government restrictions to movement during the COVID potentially prevented patients from using facilities. I therefore think that it would be useful to not attribute this to reducing access to facilities (i.e access remained the same in where facilities were not increased in number) but rather less people were likely to use facilities. Thus, I would suggest rephrasing the conclusion accordingly. For example the conclusion states the following;

“These results capture significant changes in healthcare accessibility as the COVID-19 pandemic unfolded, and indicate that the impacts to healthcare access were not distributed equally across populations within countries”

But should show that restrictions acted as a barrier to utilization of facilities, potentially even in areas with access

To a statement that captures the fact that conceptually, during covid government imposed travel restrictions reduced made people less likely to use facilities and this inturn lead to less people being captured in the realised travel.

However, few systematic efforts have characterized healthcare accessibility at the global scale and none have empirically assessed how this key public health metric has been impacted by the pandemic.

Response: We have edited the conclusion to remove the stronger conclusion on cause/effect and barriers (which as the reviewer points out needs additional work) and instead focus on describing the unequal distribution of the changes; it now reads:

“These results capture significant changes in healthcare accessibility as the COVID-19 pandemic unfolded, and indicate that **these changes** were not distributed equally across populations within countries.”

Regarding the other sentence, we have also added the travel restrictions point suggested above:

“COVID-19 pandemic has challenged the capacity of healthcare systems across the globe, **and along with travel restrictions**, potentially creating new barriers to healthcare.”

R1.3: Methods. The authors state the below;

“To identify the location of the nearest medical facility for all populated areas, we reduced gridded Landscan worldwide population density data into polygons with an average area of five square kilometers, computed the geographic centroid of each polygon, calculated the distance to all facilities nearby, and selected the closest one.”

1. What are the chances that there are multiple facilities within a 5 square km grid and this would definitely affect the accessibility metric

Response: Thank you for the question. When reviewing this, we realized that the section’s purpose was unclear based on the header; this section pertains to the health facilities inventory, and the section has been updated to state “Health facilities database”.

Similarly, whereas the first half of this section relates to both revealed and potential accessibility, the latter half (mentioned above) is only relevant to potential accessibility (where we use population density and the nearest facility to estimate potential accessibility). We have edited the beginning of this sentence to: “For potential accessibility computation, to identify the location of...”

Hence the reviewer’s question pertains to the comparison method (potential accessibility) but not to the main focus of the paper (revealed accessibility). That said, to answer the question, the situation is fairly uncommon: of the cells containing any facilities, about 18% of cells with any facilities contain more than 1 facility. When expressed as a fraction of all S2 cells of the studied size (5 square km), this is about 0.02%

R1.4: 2. In most low-income countries, access to smartphones is poor and therefore using such a method proposed would mean the results are skewed. I would suggest putting it across that this work is only beneficial in settings where access to smartphones is high and also they have location services enabled

“Briefly, for each geographic region, time period, and mode of transportation, the computation of potential accessibility involves first sampling a single trip per user to remove the effect of outliers with many trips (these trips are visible in users’ Google Maps app, under the “timeline” feature).”

Response: This is an important point. We have covered this in the Supplementary section titled “Representativeness of location data”, and added a sentence to the beginning of section containing the text above:

“Although smartphone ownership is high globally at more than 80%, we note that this distribution is not equal across several factors (see Supplementary Material).”

The relevant section is also slightly edited and copied below for convenience:

“We first note that the representativeness of location data is difficult to analyze precisely because of the need to respect user privacy. However, pertinent aggregate insights are available from reports by the Pew Research Center¹¹ among others. Specifically regarding urbanization, smartphone ownership ranges from 80% to 89% across rural, suburban and urban U.S. adults, and with respect to income levels, rise from 76% for <\$30K, to 96% for >\$75K — indicating fairly high penetration at all income levels across both urban and rural areas. Though more of a hypothesis, we believe that the skew with respect to income or urban location is likely to over-represent urban areas that have relatively better access to healthcare facilities, and thus under-estimate the true revealed accessibility.

Globally, latest estimates of smartphone penetration are high at 83%,¹⁴ and promisingly, there are signs that the “digital divide” with respect to age, such as smartphone ownership and other tech use is shrinking.¹⁵ However, some skew is likely to remain in the foreseeable future, whether for historical data or newly collected data going forward. As noted in our manuscript, caution is warranted in interpretation of the results of the study.

Finally, regarding users with the Google location history feature enabled, a 2018 study surveyed more than 1000 users across 5 countries.¹⁶ While many (20-50% of users) did not know their location history feature status, the vast majority of the remaining users had location history enabled (roughly 8:1 ratio of enabled:disabled for Japan, 3:1 for UK, 14:1 for Brazil, 5:1 for US, and 7:1 for Mexico). The authors concluded that (passive) location history data, when appropriately anonymized, was a novel and valuable way to study spatial movement while avoiding issues such as recall biases from self-report, or insufficient granularity from “CDR” (cell towers).”

R1.5 3. Sentence is not very clear and why the sampling was done. Does it mean a single user had many trips from residence to facility and therefore one was taken? Need to clarify the statement.

Response: We have clarified this to say:

“...remove the effect of outliers **with many trips such as employees or repeated visits for multiple care episodes..**”

R1.6 4. Methods needs a bit more explanation rather than packing into supplementary file. For example no information is provided on how it was possible to distinguish the use of public transport from personal vehicles and this would have an impact on accessibility estimates

Response: Thank you for the suggestion. This manuscript was originally formatted using the generic Nature Research format (Methods section after Discussion), but we believe that if accepted, Nature Communications Medicine may request a shift of the Methods to be before Results instead of after Discussion, as well as a Methods shift, so some Methods information will become more prominent.

Regarding the question on mode of transit and in particular personal vehicle vs. public transportation, this is a good question. The details are publicly described in the patent # US20170347237A1, “Determining Semantic Travel Modes”. Briefly, several kinds of metadata come in useful for this, most importantly whether segments started and ended at a public transit station. Users can see information about their mode of transit in their Timeline feature (<https://support.google.com/maps/answer/6258979>).

We have added a statement to the main Methods: “The mode of public transportation is estimated using information such as whether segments started and ended at public transit stops (e.g., bus stops or subway stations).” and cited the relevant patent above.

R1.7 5. I think the Fig A3 on revealed travel time in the supplementary file should be in the main manuscript. Also Would suggest grouping by smartphone usage/availability in countries for easier comparison and understanding of the error bars.

Response: We thank the reviewer for the Fig A3 suggestion and agree that it would be nice to have public transit and walking distances in the same figure as panels B and C.

However, given the size of the current Figure 1, we’re unsure whether this becomes too large for the journal format. We defer to the editor on any style preferences, and would be happy to shift the figures if desired.

Regarding the smartphone usage suggestion, this information is unfortunately not always available. Since smartphone ownership is correlated with income levels, we have instead re-plotted Figure 1 with colors indicating the income level of the country (as categorized by the world bank). This is now indicated in the caption.

As the reviewer intuited, this new visual shows that the longer travel times are more common in lower income countries, and we appreciate the suggestion that has helped make this clearer.

Reviewer #2 (Remarks to the Author):

R2.1: Does the manuscript have technical or conceptual flaws that should prohibit its publication? If so, please provide details.

No, but I feel this research could be better located within the existing literature, particularly more recent studies on measuring accessibility to health care (in general or specific services) and the impact of the pandemic in access/availability of health services (very little literature on this is cited or explored). I would also suggest that justification of the choice of datasets is needed, and explanation of which countries are considered in the analysis (particularly as there are many references to the analysis being global in nature).

Response: Thank you for raising this. We have added more than 10 recent references. The new additions in the text are:

Discussion paragraph 1: “ Our findings are consistent with previous work demonstrating that spatial healthcare access estimates within a country can vary substantially depending on the estimation method ^{32,33,34,40} .”

Discussion paragraph 2: “However, the application of existing methods has been limited to specific countries and specific healthcare services ^{13,35,36,41} , low-resource settings ^{37,39} , or only capture certain dimensions of potential accessibility worldwide.¹ Similarly, geo-spatial studies aiming to measure healthcare access during COVID-19 pandemic have largely been limited to specific countries and relying on potential accessibility-based catchment-area methods ³⁸ .”

Discussion paragraph 3: “Our results echo previously identified knowledge gaps and calls for more fine grained and temporally-aware accessibility metrics, more sophisticated geocomputational tools to operationalize such metrics, and improved measurement of inequalities ⁴² .”

Regarding the choice of dataset, we have added to the main text: “using passive anonymized data helps avoid issues with alternative methods such as recall bias with self-report or insufficient granularity with cell tower data.” and cited:

Ruktanonchai, N. W., Ruktanonchai, C. W., Floyd, J. R. & Tatem, A. J. Using Google Location History data to quantify fine-scale human mobility. *Int. J. Health Geogr.* 17, 28 (2018).

Regarding which countries are included in the analysis, the full list of 120 countries is in Supplementary materials section A.6, and the list of countries is also in Figure 1. We have added a reference to this Figure in the introduction.

Regarding the choice of countries, we clarified it at the beginning of the results (at the first mention of the numbers “193” and “120”, as follows: “, *based on the availability of reliable data (Methods).*”). We have also ensured that the Supplementary discussion on the details of differential privacy and filtering are present and referred to from the Methods (“More details on this step and filtering for sufficient data are presented in Supplementary Materials.”).

R2.2: Are the conclusions original? If not, please provide relevant references.

The analysis provides new insights into potential and “realised” access to health services, considering both a pre-pandemic and during pandemic time period, across many countries. I would question whether the paper’s claims of being useful for understanding where to locate new facilities and services, are really the case, as the results presented are all at national level – if the analysis provides sub-national results, it would be interesting to include these also.

Response: This is an important point, our analyses are currently performed at the country level. Although the fundamental methods will likely be similar, future work in collaboration with local stakeholders will be needed to focus on specific sub-national locales. We see how the previous wording in the Discussion could be overly broad and have rephrased it to:

“...Future work is needed to apply these methods at regional and metropolitan level, in order to make the resulting insights actionable to local stakeholders. For example, health planners could consider revealed travel times in planning for healthcare service provision, optimizing public transit routes, or spatially targeting health promotion campaigns....”

R2.3 Do you feel that the results presented are of immediate relevance for many people in your own field or for a broader audience?

I think the results will be of interest to a wide audience, however the presentation of results being limited to the national level will likely limit its interest for those actually involved in planning health resource allocation at a more local level. The limited explanation of which countries are included in the analysis also makes it difficult to interpret how widely the results are applicable.

If applicable, does the manuscript reporting follow ICMJE and EQUATOR recommendations? If not, please provide details.

Not applicable

If you recommend publication, please outline briefly what you consider to be the outstanding features.

If the issues outlined in these comments are addressed satisfactorily, I would recommend publication – most relate to the justification of datasets used, the time periods/countries considered and the situating this research better within the existing literature. This paper's main strength is the use of Google datasets that are not widely available to researchers, and the new insights that this presents, across multiple countries.

Response: Regarding which countries are included in the analysis, the full list of 120 countries is in Supplementary materials section A.6, and the list of countries is also in Figure 1. We have added a reference to this Figure in the introduction at the first mention of “global level”.

Regarding the time periods, we have verified that the time is listed in Methods as “time period (calendar year quarters, Q1 2019 - Q3 2021)”, and where applicable the time periods associated with changes are reported in the caption (Table 1: Q2 2020 vs. Q2 2019; Figure 2: time is on the x-axis; Figure 6: Jan 2019 - Dec 2019). Where missing, we have added these for clarity (Figure 2 and Figure 3).

R2.4 If you feel that specific additional experiments would strengthen the case for publication in Communications Medicine, please provide suggestions.

As described above, inclusion of analysis for sub-national locations would strengthen this paper.

Response: We thank the reviewer for the suggestion and refer the reviewer to our answer to question R2.2 (by the same reviewer) above. Briefly, we have clarified in the Discussion that the sub-national analysis is part of future work and likely needs to be done in collaboration with the public sector of specific jurisdictions.

R2.5: General comments

Thank you for the opportunity to review this paper. It is an interesting analysis utilising Google location history data across many countries, and Google routing applications to calculate potential travel times. This is explored in two time periods: just before and during the COVID-19 pandemic. The following are general comments which I hope can help make the analysis and interpretation of results clearer to readers.

Response: We appreciate the thoughtful suggestions and positive comments by the reviewer.

R2.6: • Please clarify if accessibility is being measured to all healthcare facilities (defined how?) or only a subset offering. In the fourth paragraph of the introduction, you state “the inventory of geolocated hospitals and medical centers providing urgent and emergency care...” which suggests it’s just a subset of health facilities being considered, but throughout the rest of the paper, you make reference to healthcare, healthcare facility and medical facility. Then in the limitations, you write “because not all facilities offer all services, cannot be used to assess travel times to specific services”, which seems to contradict your initial statement. It would help those reading the paper if you make this clear early on in the paper and then use consistent terminology throughout.

Response: We understand that this wasn’t clear enough and have edited the first Methods header to “Medical Facility Database” to be clearer, and also added a reference to this section in the Introduction where “inventory” was first mentioned.

As mentioned above, Methods will also be shifted to before Results if accepted at Communications Medicine, which will further emphasize this section.

Regarding our limitations comment on travel to specific services, we have rephrased it be clearer: “Fourth, this analysis was a general examination of healthcare access across many facilities, and because information on facility-level service offerings were not available, this analysis does not provide granular insights about travel times to specific services such as primary care or specialty care services.

R2.7: • At the end of the abstract you state that the results can be used by policy makers for directing resources and formulating policies – suggest indicating that these results should be considered in conjunction with other sources, i.e. not used in isolation.

Response: This is a good point and we have rewritten the text as follows: “Public health policymakers may use these results in conjunction with other relevant data for formulating policies and directing resources towards areas and populations most in need.”

R2.8: • The geographic scope of the paper should be clarified. There is mention of this being a global analysis, but then at the start of the results then you state “we quantify population-weighted potential and revealed travel times to healthcare in 193 and 120 countries respectively”, without explanation of why and which countries have been included in this analysis. I presume this is probably due to data coverage, but this should be clearly stated. Could the inclusion of only a subset of countries potentially alter results compared to a truly global analysis?

Response: Thank you. We have edited the text to state: “We quantify population-weighted potential and revealed travel times to healthcare in 193 and 120 countries, respectively, based on the availability of reliable data (Methods).”

Regarding whether these results may change, with a total of 195 countries in the world, the potential travel times above are fairly comprehensive. For revealed accessibility, we have plotted the geographical coverage of the 120 countries:

Based on 2019 world bank data, the 120 countries include a population of 5.5 billion, which is approximately 71% of the global population (87% after excluding China).

Finally, to further address this, we have edited all mentions of global in the context of the study's scope, to be more specific that this deals with "more than 100 countries".

R2.9: • Did you do any analysis to understand the coverage of the health facility data used in comparison to other sources e.g. OSM or WHO health facility list (Africa only)?

Response: We agree that this is important and have indeed conducted this analysis. This is covered in our limitations section: "First, our inventory of healthcare facilities (based on Google Maps) may be incomplete or inaccurate, with the quality and completeness of the dataset likely to vary between countries (Supplementary Material, Table A.4)".

The relevant Supplementary section (A.4 The inventory of healthcare providers) details the breakdown of the overlap of healthcare facilities between our datasets and healthsites.io, at both the global level and when stratified by country-level income categorization.

R2.10: • A few comments related to figures:

- o Figure 3 – the world maps are too small to see any changes for small countries. Please also indicate what grey is (no data?). Also consider a colour scale where 0 is not the same colour as the background/non-landmass
- o Figure 3 caption – consider stating the time period for "pre-pandemic" and "COVID-19 pandemic"
- o Figure 4 – the crowded labelling of data points in a, b and c makes this very difficult to read or interpret. Please revisit

Response: Thank you for the thorough review. We've made the following edits:

Figure 3: caption edited to add the baseline and COVID period (2019 vs 2020-2021, respectively), and to describe that gray is lack of data. The colors have also been updated such that the background is blue and the land is green. We have also increased the resolution to enable readers to zoom into individual countries better, and defer to the journal on style preferences.

Figure 4: we understand the reviewer's concerns and have tried a few different ways of plotting in response to this suggestion. Unfortunately it is challenging to individually indicate all (over 100) countries in the plot while enabling every word to be legible and minimizing overlaps. We have attached 3 high resolution PDF files for panels a-c where this crowding is most apparent, to enable users to dive in more deeply. In addition, we hope that the outliers (which are not affected by crowding) such as Zambia are legible and interpretable (ie, having both poor health outcomes and high revealed travel times).

R2.11: • In the limitations section you ask whether users are a representative sample of the general population – what does the literature indicate about this?

Response: This is an important point. A more in-depth discussion is in the Supplementary information, section "A.2 Representativeness of location data".

We have copied the most relevant section discussing "users" here:

"Finally, regarding users with the Google location history feature enabled, a 2018 study surveyed more than 1000 users across 5 countries.¹⁶ While many (20-50% of users) did not know their location history feature status, the vast majority of the remaining users had location history enabled (roughly 8:1 ratio of enabled:disabled for Japan, 3:1 for UK, 14:1 for Brazil, 5:1 for US, and 7:1 for Mexico). The authors concluded that (passive) location history data, when appropriately anonymized, was a novel and valuable way to study spatial movement while avoiding issues such as recall biases from self-report, or insufficient granularity from "CDR" (cell towers)."

To make this clearer, we have edited the Limitations paragraph to have a clearer pointer: "As such, the results of this work should be interpreted cautiously in cases where the movement patterns of a population of interest may differ from those in our study population. We further discuss this issue and data from the literature in the Supplementary Material."

R2.12: • In the final paragraph of the introduction you state "we leveraged the spatial patterns of smartphone usage" – this is quite a broad statement, and would suggest a data source that represents a wider population than is captured by Google Location History. Please make this clearer.

Response: Thank you for the detailed read. We have edited this to add “ via anonymized Location History data”.

R2.13: • In the first paragraph of the results, please clarify the mode of transport for the under 44 minutes result.

Response: We have added “using a passenger vehicle”.

R2.14: • In the results section, when stating results for “during the pandemic”, please be clear the time period being referred to.

Response: We have added “(first quarter of 2020 to third quarter of 2021, with the respective quarters in 2019 as the baseline)” in the first sentence of the COVID section of Results.

R2.15: • In the third paragraph on page 4, when you discuss changes in travel time, have you considered whether socio-economic impacts may have changed the make-up of e.g. who was walking?

Response: This is a great point, that there may have been a COVID-associated shift in the population utilizing the three modes of transportation studied. Because data were anonymized at the level of country, calendar year quarters, and transportation mode (Methods), we are unable to evaluate the population composition whether pre- or during the pandemic.

We have added to the limitations: “Similarly, individuals’ socioeconomic circumstances before versus during the pandemic and choices regarding the mode of transport may have influenced the trends observed.”

R2.16: • In the final paragraph of the results (page 5), there’s mention of biggest differences in sub-Saharan Africa, which I presume is referring to the top 3 countries in figure 6 being Zambia, Rwanda and Nigeria. If this is case, consider referring to these countries by name. There are other sub-Saharan African countries ranked much lower in figure 6 (e.g. Kenya, Mali, Gabon). From a quick count there appears to be only 11 SSA countries shown in figure 6, whereas there are over 50 countries in SSA, so difficult to know if similar patterns would be seen in other countries not included in the analysis.

Response: We agree with the reviewer and have replaced the mention of SSA with a list of the top 3 country names (“Zambia, Rwanda, and Afghanistan”). The new Figure 1 also colors the countries based on World Bank data, which makes it easier to see that all of these 3 countries are low-income countries.

R2.17: • In the analysis comparing travel times and health outcomes (final paragraph page 4), it's not immediately clear at what geographic scale this comparison is being made (only at national-level I think?). And the health outcome indicators are presumably only from pre-pandemic period? I think it's important to explain this more explicitly.

Response: We have edited the first sentence of that paragraph to say "country-level correlation", and added "(from 2017-2018; see Figure 4)".

R2.18: • The Methods section is currently right at the end of the paper (after figures and references). My understanding is that Comms Med papers have the methods section integrated into the main text, so I presume this will be changed? If not, then I would suggest adding some details earlier in the paper explaining how the pre-pandemic baseline was defined and the dates it covers, similarly for the period of the COVID-19 pandemic considered. Similarly I would suggest adding an explanation of the source of population numbers and populated places earlier in the text to help readers have a clearer understanding of the analysis.

Response: We thank the reviewer for noting this and will move the Methods section as part of the editorial checklist (provided upon acceptance/provisional acceptance).

R2.19: • When comparing results from during the pandemic to pre-pandemic baseline, was there any consideration around differences in lifting of restrictions in different countries?

Response: This is an important point - that different countries had restrictions of different magnitudes and time horizons ranging from a few days to months (e.g., see https://en.wikipedia.org/wiki/COVID-19_lockdowns_by_country). However, because of this variability in duration and the fact that the differential privacy for data anonymization was done at the quarter (3-month) level, it would be difficult to tease out short time-frame patterns.

As a sanity check, taking Australia as an example (which had some of the longest country-level lockdowns at ~2 months in Aug through Oct 2021):

	Passenger vehicle	Public transit	Walking
Pre-pandemic and lockdown (Q3 2019)	60	55	27
Q3 2020	52	54	34
Quarter of lockdown (Q3 2021)	51	50	36

As may be apparent above, lockdowns likely reduced traffic, improving travel times by vehicles (passenger or public transit), but increased walking times. This is largely consistent with our observations in Figure 2.

Additionally, we will be providing the anonymized data above for all 3 modes, countries, and time periods studied, to enable others to conduct this or other similar analysis.

We have added detail to the limitations:

“Similarly, exogenous factors such as the COVID-19 pandemic also influenced the type and mode of care (e.g., preventative vs. urgent, in-person visit vs. telemedicine, lockdowns and ability to travel to the care facility), and this anonymized analysis measures the net effect on in-person travel without the ability to break down the data based on these factors.”

R2.20 • In the Methods section, the description of the processing of the LandScan data needs to be revisited and the explanation improved. When you state “we reduced gridded LandScan worldwide population density data into polygons with an average area of 5 sq. km”, I presume you are meaning that the population density raster was resampled (i.e. the grid cell size changed) and the output grid cell size was increased? Please clarify whether the resampled grid cell size was 5x5km (i.e. 25 sq. km) or actually ~2.23x2.23km (i.e. 5 sq.km) – currently you are indicating the latter, but I think you may be meaning the former?

Response: Thank you for the question. We do indeed mean 5 sq km as written, and have clarified this with additional text: “(roughly 2.23 km by 2.23 km; S2 cells level 12)”.

R2.21 • Is the above the process for defining “populated places”, if so please state this or if not please explain how populated places are defined.

Response: We have added “(with a population of at least 50 people based on Landscan data)” right after “populated places”.

R2.22 • Please justify the choice of LandScan population dataset? You may want to consider recent work by Hierink et al. (2022) which explores the impact of utilising different gridded population datasets on accessibility measures.

Response: Thank you for the pointer and the helpful reference. Hierink et al. state in their paper: “Coverage statistics are highest when using GHS-POP (95.0%), HRSL (93.2%), LandScan (93.4%), or WorldPop top-down constrained (84.3%).” Given this and the fact that travel times in their study (Figure 1 in the paper) were very divergent but converged at the higher end, it’s likely that differences narrow for the longer travel times analysis.

We have added to the Methods and cited the paper above: “Whereas Landscan has one of the highest population dataset coverage³³, exploration of other population datasets may be helpful.”

R2.23 • Much of the literature cited is relatively old (15/31 references are from 2011 or earlier). Please try to ground this work in more recent literature as there has been considerable work in the last decade on measuring accessibility to health facilities/different types of healthcare. Just a few of the many recent papers/chapters to consider:

o Ouma et al. (2021) Methods of Measuring Spatial Accessibility to Health Care in Uganda, https://doi.org/10.1007/978-3-030-63471-1_6

o Hierink et al. (2022) Differences between gridded population data impact measures of geographic access to healthcare in sub-Saharan Africa, <https://doi.org/10.1038/s43856-022-00179-4>

o Bouanchaud et al. (2022) Comparing modelled with self-reported travel time and the used versus the nearest facility: modelling geographic accessibility to family planning outlets in Kenya, doi:10.1136/bmjgh-2021-008366

o Dotse-Gborgbortsi et al. (2020) The influence of distance and quality on utilisation of birthing services at health facilities in Eastern Region, Ghana, doi:10.1136/bmjgh-2019-002020

o Wigley et al. (2020) Measuring the availability and geographical accessibility of maternal health services across sub-Saharan Africa, <https://doi.org/10.1186/s12916-020-01707-6>

Response: Thank you for raising this. We have added more than 10 recent references (including the 5 above). The newly added text are:

Discussion paragraph 1: “ Our findings are consistent with previous work demonstrating that spatial healthcare access estimates within a country can vary substantially depending on the estimation method^{32,33,34,40} .”

Discussion paragraph 2: “However, the application of existing methods has been limited to specific countries and specific healthcare services^{13,35,36,41}, low-resource settings^{37,39}, or only capture certain dimensions of potential accessibility worldwide.¹ Similarly, geo-spatial studies aiming to measure healthcare access during COVID-19 pandemic have largely been limited to specific countries and relying on potential accessibility-based catchment-area methods³⁸ .”

Discussion paragraph 3: “Our results echo previously identified knowledge gaps and calls for more fine grained and temporally-aware accessibility metrics, more sophisticated geocomputational tools to operationalize such metrics, and improved measurement of inequalities⁴² .”

Reviewer #3 (Remarks to the Author):

R3.1: General Comments.

The manuscript presents a novel analysis of the impact of COVID on geographic accessibility, comparing hypothetical estimates with actual estimates using Google data. However, the validity of using smartphone data to track movement for global comparisons is a concern, due to unequal distribution of smartphones, which limits the representativeness of the results. The authors acknowledge this limitation but further clarification is needed to justify the use of global comparisons. Additionally, the exclusion of information from several countries raises questions about the accuracy of the global analysis.

Response: We thank the reviewer for comments on the novelty of the work.

We also appreciate the thoughtful comment on limitations regarding drawing global inferences at the present time. We have edited the only mention of 'global' in the abstract to "in over 100 countries", and also edited other mentions of the word "global" accordingly. In the Results, we have also defined "global" at first mention to be clearer that the data refers to the 193 and 120 countries studied.

R3.2: Reviewer comments:

1. Introduction:

- The focus of the work should be revised to reflect that it is not solely on comparison between pre- and post-pandemic comparison.

Response: We appreciate this being pointed out and have edited the title:

"Revealed versus Potential Spatial Accessibility of Healthcare and Changing Patterns during the COVID-19 pandemic"

And in the Introduction, we have ensured the final paragraph is clearer on the sequence of work:

"This paper reports a novel approach to the development of measures of both potential and revealed access to care in over 100 countries (Figure 1)... We estimated the travel time to medical facilities starting from populated places by car, public transport, and walking. Finally, we analyze how travel time has changed during the COVID-19 pandemic...."

R3.3: • The introduction needs to be more focused on Geographic access and the comparison between modelled and actual travel times/distance.

Response: We acknowledge that our work's contributions can be made clearer and have made the edits in the response immediately above (see R3.2).

R3.4: • The introduction speaks a lot about the broader concept of health access in general and not enough background to the actual work presented.

Response: We provided additional general background in paragraphs 1-3 in the introduction to help the broad readership of Nature Research journals. As explained in item R3.2 above, we have clarified the work in paragraph 4 of the introduction.

R3.5: • Include more information on the importance of comparing potential vs revealed access or the discussion underpinning the hypothesis around why COVID would affect access needs to be explained.

Response: We have added the following to Introduction paragraph 3 (which introduces potential and revealed accessibility):

“Whereas potential (theoretical) accessibility based on routes and population density is helpful in planning, it is important to understand if reality (revealed accessibility) deviates from theory, and by how much. Understanding this difference and how it is influenced by crises (such as the COVID-19 pandemic) can help further refine planning and catalyze efforts to improve healthcare accessibility.”

R3.6: 2. Focus of the work:

- It is unclear what the focus of the work is (comparison between potential and revealed or how geographic access has changed over time in relation to COVID).

Response: As the reviewer notes, we have presented analyses on both potential vs. revealed accessibility and changes during COVID -- but the title did not reflect this. We have edited the title to encompass both goals:

“Revealed versus Potential Spatial Accessibility of Healthcare and Changing Patterns during the COVID-19 pandemic”

R3.7: 3. Changes in transport modes:

- The fact that people may have stopped using public transport in favor of private vehicles to reduce exposure or the fact that public transport was stopped in some countries needs to be accounted for.

Response: We acknowledge that shutdowns or lockdowns were implemented in many countries during COVID, with their duration ranging from a few days to months (eg, see https://en.wikipedia.org/wiki/COVID-19_lockdowns_by_country). However, because of this

variability in duration and the fact that the differential privacy for data anonymization was done at the quarter (3-month) level, it would be difficult to tease out short time-frame patterns. However, we will be providing the anonymized data to enable others to conduct this or other similar analysis.

We have added detail to the limitations:

“Similarly, exogenous factors such as the COVID-19 pandemic also influenced the type and mode of care (e.g., preventative vs. urgent, in-person visit vs. telemedicine, lockdowns and ability to travel to the care facility), and this anonymized analysis measures the net effect on in-person travel without the ability to break down the data based on these factors.”

R3.8: • COVID restrictions varied across countries and over time which may influence broad statements around

Response: This appears to be a truncated sentence. If we understand correctly, the reviewer is pointing out that some impacts of COVID cannot be modeled using this approach and temporal granularity. We have verified that our language around changes during COVID does not make causal statements around the source of the COVID-related changes, but rather focuses on the changes themselves: “These results capture significant changes in healthcare accessibility as the COVID-19 pandemic unfolded...”

Please also see the point immediately above this one (R3.7) for how we more explicitly spelled out limitations around COVID.

R3.9: 4. Results:

- The inclusion of other factors like telemedicine and delayed elective procedures and their impact on mode-specific travel time needs to be explained. Presumably these don't contribute to the travel time metric because they are non-occurrences

Response: This is a great point - that non-occurrences (utilizing telemedicine or postponing care instead of traveling in person) would not be measured by our approach because the trips did not happen. We have outlined this similarly in the response above, specifically:

“... (e.g., preventative vs. urgent, in-person visit vs. telemedicine, lockdowns and ability to travel to the care facility), and this anonymized analysis measures the net effect on in-person travel...”

R3.10: • The value of including the comparison of revealed travel time to health outcomes is unclear, especially because of the estimates of health outcomes in from 2017 and especially concerning the comparison in figure 4d.

- Also, I am not sure this piece of analysis adds to the work presented

Response: We understand the comment and have edited the Results to be clearer about the goal (country-level correlation) and the years the outcome information are from:

“We observed a strong country-level correlation between the revealed travel time to healthcare facilities and quality of health outcomes (from 2017-2018; see Figure 4).”

We have also added a pointer to the Supplementary multivariable analysis that models outcomes and both potential and revealed accessibility at the same time:

“Multivariable analyses incorporating travel times and GDP are presented in Supplementary Section A.9, with both revealed travel times and potential travel times remaining significant after adjusting for GDP.”

Finally, we have edited the discussion section to clarify the takeaway:

“Finally, we show that revealed accessibility correlates with important health outcomes across countries, even after adjusting for potential accessibility and wealth (GDP), indicating that accessibility is an important independent factor to track, understand, and improve.”

R3.11: 5. Distinguishing between a passenger vehicle and public transport:

- Define what is meant by passenger vehicle vs public transport. Public transport also include passenger vehicle
- The method used to distinguish between a passenger vehicle and public transport, especially in countries where passenger vehicles are similar to public transport, needs to be made clear.

Response: Indeed there does not seem to be a single widely-accepted definition of “passenger vehicle” (definitions range from being based on vehicle weight or length, to excluding categories such as buses/planes, to adding modifier words such as commercial when referring to public transportation). Though “passenger vehicle” is purportedly the US Department of Transportation’s term for cars and other private vehicles (https://en.wikipedia.org/wiki/Passenger_vehicles_in_the_United_States), we were unable to find the precise definition on .gov websites.

Given that this study involves multiple countries, we agree with the reviewer that it is prudent to define this properly. To the first paragraph of Methods, we have added:

“For the purpose of this study, passenger vehicles are defined as vehicle travel excluding public transportation (buses, train, subway, etc), and the distinction in trips is made using public transportation station information (see below).”

R3.12: 6. Health facility database:

- The databases used for the health facility database need to be specified and completeness will vary geographically and between developed and developing countries. This needs to be acknowledged.

- The data extraction method (from Google maps and search datasets) needs to be explained. There are several available health facility databases available why restrict this to a single comparison.
- Indicate the temporal timepoint reference for health facility databases used

Response: We have ensured that the limitations section contains this acknowledgement: “First, our inventory of healthcare facilities (based on Google Maps) may be incomplete or inaccurate, with the quality and completeness of the dataset likely to vary between countries (Supplementary Material, Table A.4).”

Further, we have edited the Methods to more clearly point to the supplementary analysis: “We compared this inventory with publicly available inventories of medical facilities; coverage varies across countries (supplemental material)”

Regarding the data extraction method, this is an internal dataset, but one that everyone can access by searching “healthcare facility” or similar on Google Maps or Search. We have added this statement to the renamed section “Medical Facility Database”.

Regarding the reason for using this Google dataset, the novelty of this study involves measuring revealed accessibility via anonymized trip travel time information. As such, we needed to ensure consistency between the method of defining trips and the facility location information. Unfortunately, mapping between facility datasets is challenging, with many facility lists lacking consistent geographical coordinates or street address information (e.g., some are based on directions and word-of-mouth). We have added the following under Methods:

“...this dataset was used to ensure consistency with anonymized trip location information, described below.”

Regarding the timepoint, we have added “ in August 2019”.

R3.13: 7. Accessibility modeling:

- More detail needs to be provided on how the accessibility modeling was done.
- Was the computation of revealed accessibility restricted to country borders?
- The countries from which the analysis was able to be run needs to be indicated in the main text methods and why some countries only had information on PV and not for public transport
- Clarify what Google Maps Directions API is for non-familiar users

Response: We have clarified that revealed accessibility estimation (i.e., the definition of trips that underwent aggregation/anonymization) focused on trips within countries only.

We have edited the Methods to say “Trips are constrained to within-country trips only.”

Regarding the list of countries, we have added a pointer to the requested list of countries (presented together with the per-country revealed accessibility) to Figure 1: “Data for other modes of transportation are presented in Figure A.3.”

Regarding the Google Maps API, we have added: “We estimated potential travel time from every location to the nearest healthcare facility using the Google Maps Directions API, which is an automated service that returns travel directions based on input such as a requested origin, destination, and mode of transportation”

R3.14: 8. Figure 3:

- The purpose of the grey areas in Figure 3 needs to be explained.
- The countries for which the authors had data and the consistency of this data across different modes of travel needs to be mentioned in the methods.

Response: We have added to the caption: “Gray color indicates countries or regions not included in the study; see Figures 1 and Supplementary Section A.6 for the list of studied countries.”

We have also added to Methods: “The exact list of countries studied for each mode of transportation is listed in Figures 1 and Supplementary Section A.6.”

R3.15: 9. Figure 5:

- The county estimates shown in Figure 5 need to be specified.
- The missing data on public transport needs to be explained.

Response: We believe these questions are related to the point immediately above (R3.14) and they are addressed in the Methods and Figure caption addition. In addition to above, we have also edited Methods to state: “Driving and walking potential travel time estimates could be computed for 193 regions, whereas public transportation information was not available for some regions in Google Maps and was estimated for 91 regions.”

R3.16: 10. Use of smartphone data:

- Concerns around the use of smartphone data for revealed travel time (as only 46% of populations in SSA have a smartphone and an even smaller percentage have location enabled) need to be addressed.
- The potential for non-representative populations and problematic cross-country/cross-regional comparisons need to be expanded on in the limitations. It potentially compromises the comparisons presented in the paper
- Hence wonder if it makes more sense to present this data by regions

Response: Thank you for the thoughtful comments. We have revised as above to be more specific that the analysis reflects a hundred countries instead of every country globally. The limitations section has also been revised to state: “Third, we excluded from the analysis

countries with a low number of trips to healthcare facilities, so additional countries will need to be covered in future work as smartphone ownership and the coverage of healthcare facilities increases.”

Regarding the comparison by regions, this may be important follow-up work by teams interested in the specifics of their countries and comparisons with neighboring comparable countries. In addition to the limitations revision above, we are providing the aggregated data to enable others to dive into these analyses.

Reviewer #4 (Remarks to the Author):

R4.1: I congratulate the authors for the impressive and innovative work that they present. The paper is novel and presents the use of new technologies in the field of geographical accessibility modelling. The comparison between potential and revealed travel time adds new realism to accessibility modelling and reveals important differences. The effect of COVID-19 on revealed travel times to health facilities shines light on the impact of large-scale health emergencies on health seeking behavior and challenges in accessing care. I have a few concerns regarding the paper that I would like to share in the points listed below and I encourage the authors to address these to further improve the manuscript. I remain at the authors disposal in case they have any questions considering the below points.

1. The authors state that countries with a low number of trips to health facilities were excluded. However, the authors do not state what number was considered as the threshold and how many users were analyzed/included per country. The authors should state what number of users was considered representable and reliable and why. The number of users may also vary a lot between countries and therefore largely impact results.

Response: We thank the reviewer for the positive comments.

Regarding the exclusion threshold, we have verified that this is stated in the supplementary information (“threshold of $k = 1000$ ”), and have edited the main Methods to ensure this pointer is clearer: “More details on this step and filtering for sufficient data are presented in Supplementary Materials.”

In addition, upon acceptance, this Methods-last format will be edited such that this information becomes clearer.

Regarding the number of trips, this is a good question. Whereas 1000 was arrived at qualitatively with considerations towards user privacy, we can also approach this from a statistical perspective. The analyses focus on quantiles (10th, 25th, 50th, 75th and 90th percentile), and the confidence intervals for quantiles can be derived analytically or using a binomial approximation (<https://online.stat.psu.edu/stat415/book/export/html/835>).

Computing the amount of probability mass within 2 percent of each quantile (as a measure of dispersion and confidence in the quantile estimates), we obtain for 90th percentile:

n	CDF (quantile-2)	CDF (quantile+2)	Probability mass in +/- 2
100	0.25	0.75	0.50
500	0.07	0.93	0.86
1000	0.02	0.98	0.96
5000	0.00	1.00	1.00
10000	0.00	1.00	1.00

And for 50th percentile or median (confidence intervals are larger when the quantile is closer to the median):

n	CDF (quantile-2)	CDF (quantile+2)	Probability mass in +/- 2
100	0.34	0.66	0.31
500	0.19	0.81	0.63
1000	0.10	0.90	0.79
5000	0.00	1.00	1.00
10000	0.00	1.00	1.00

Thus, the confidence in the quantile estimates concentrates quickly when sample sizes exceed 1000, improving statistical reliability of estimates and comparisons. We have added to the section discussing this threshold:

“This threshold of $k=1000$ was determined qualitatively for privacy reasons, but ensuring that a sufficient number of trips are incorporated in each estimate also serves to improve the certainty in each estimate. For example, using a normal approximation, for the 90th percentile, with 1000 trips the 95% confidence interval is small at approximately +/- 2 percentiles.”

R4.2: 2. The authors state a high inequality in revealed accessibility in several low- and middle-income countries (e.g. Zambia, Rwanda, Afghanistan, Zimbabwe and Nigeria). I think that much of the variation found in countries in sub-Saharan Africa and Central Asia is impacted by a higher number of users in urban areas and a low number of users in remote areas, potentially causing outliers to have a greater effect. This will likely cause even larger variation between the percentiles than can be already expected. Did the authors consider making any adjustments or corrections for the number of users in the different regions of a country? Or distinguishing urban/rural differences?

Response: We agree that the number of users has an impact on statistical precision and refer to the answer immediately above on the overall effects. Regarding urban vs. rural specifically, this analysis was done at the country level to preserve privacy, and so this breakdown is not available.

However, we are happy to follow the reviewer's thought experiment to understand the impact of lacking rural users. If the trips data are enriched for urban inhabitants (beyond population-level urban-rural skew; the World Bank estimates that 56% of the global population lives in cities), then our estimates are under-representing the rural inhabitants. If we assume (as the reviewer suggests, and we agree) that rural inhabitants have to travel farther to find care, then lacking these longer trips means that our estimates are under-estimating revealed and potential accessibility -- and the true inequality will be even greater than reported here.

We have added this to the limitations as " In turn, if phone ownership is more common in urban environments where travel times are shorter, the inequities observed in this study may constitute underestimates of the true state."

R4.3: 3. All of the results presented in the main manuscript and the supplementary material are on a national scale. The aggregation to national level complicates the identification of sub-national variation and thus gaps in service delivery which are extremely important. I encourage the authors to present some of the results sub-nationally or to elaborate why results could only be presented at national scales.

Response: We acknowledge that this work focuses on the country level owing to the approved granularity of the work.

We fully agree that sub-national level analysis is key for local officials, and expect that such fine-grained analyses will need to be conducted in collaboration with the respective public offices, and with more specific approvals and data on specific local regions and timespans of interest. We have updated the limitations section to state: "Similarly specific analyses on the subnational level such as urban vs. rural differences or local regions or time spans of interest will need to be explored in future work and in collaboration with local public offices"

R4.4: 4. Another important aspect is the change in users during the pandemic. Considering that many of the high-income countries and even some low-and middle-income countries had lockdowns in place, preventing people to go outside and thus having to use Google Maps or other related services. Was there a change in users and if so, what is the impact of the change in the number of users? How do the authors ensure that this doesn't impact their results?

Response: We agree that lockdowns are important to study. Unfortunately a challenge is that different countries had restrictions of different magnitudes and time horizons ranging from a few days to months (eg, see https://en.wikipedia.org/wiki/COVID-19_lockdowns_by_country). Because of this variability in duration and the fact that the differential privacy for data

anonymization was done at the quarter (3-month) level, it would be difficult to tease out short time-frame patterns.

As a sanity check, taking Australia as an example (which had some of the longest lockdowns in several regions recorded in the link above at ~2 months in Aug through Oct 2021):

	Passenger vehicle	Public transit	Walking
Pre-pandemic and lockdown (Q3 2019)	60	55	27
Q3 2020	52	54	34
Quarter of lockdown (Q3 2021)	51	50	36

As may be apparent above, lockdowns likely reduced traffic, improving travel times by vehicles (passenger or public transit), but increased walking times. This is largely consistent with our observations in Figure 2.

However, we will be providing the anonymized data to enable others to conduct this or other similar analysis.

We have added detail to the limitations:

“Similarly, exogenous factors such as the COVID-19 pandemic also influenced the type and mode of care (e.g., preventative vs. urgent, in-person visit vs. telemedicine, lockdowns and ability to travel to the care facility), and this anonymized analysis measures the net effect on in-person travel without the ability to break down the data based on these factors.”

Regarding whether the number of users specifically affects these results, we direct the reviewer to their first comment (R4.1), where we provide data on the statistical properties of quantiles and how mass concentrates with sample size. Briefly, above 1000 trips, the statistical precision of the estimates become higher.

R4.5: 5. Many hospitals during the lockdown remained closed for visitors. How did the authors make sure to factor-in a potential difference in the noise of Google users going to other places falling within the 500m radius?

Response: Please also see the answer immediately above (R4.4) on lockdowns and their shorter duration than the time scales in this work (days/weeks vs. 3 months). With that caveat, it was unfortunately not possible to account for given user privacy considerations. We have ensured that the limitations spells this out: “Second, to preserve user privacy, we used a radius of 500m around each facility, which potentially included some non-healthcare-seeking trips in the analysis (the analysis may also include trips by people who visit hospital patients but do not receive healthcare themselves).”

R4.6: 6. The authors present the impact of revealed travel times on health outcomes during the lockdown. Yet, the manuscript does not describe where this health outcome data comes from, nor how it was analyzed. The authors should describe in more detail how this analysis was done and with what data if they decide to present these results. These health outcomes might not only be attributable to travel times, but also to quality of care, availability of staff, etc. during the pandemic. This should be mentioned with more emphasis.

Response: We appreciate the comment and have updated the Methods to include this: “We correlated our measures of accessibility and inequality with published metrics reflecting health outcomes within populations, specifically national infant mortality rates and life expectancies from the World Bank”.

Specifically regarding impact on outcomes, it appears we had inadvertently suggested these outcomes were COVID-associated. The Figure axes labels and captions are now clearer that these outcome information were pre-pandemic (2017-2018). Results are also clearer now about this point: “We observed a strong country-level correlation between the revealed travel time to healthcare facilities and quality of health outcomes (from 2017-2018; see Figure 4).”

Finally, we have updated the Discussion to now state the goal of country-level analysis more clearly (“Finally, we show that revealed accessibility correlates with important health outcomes across countries, even after adjusting for potential accessibility and wealth (GDP), indicating that accessibility is an important independent factor to track, understand, and improve.”)

R4.7: 7. The health facilities included in the analysis were fetched using Google maps and a comparison was made with healthsites.io. Even though these platforms provide useful data in the absence of government data or other reliable data sources, there are often large discrepancies between the different facilities lists (South et al., 2021. <https://wellcomeopenresearch.org/articles/5-157/v1>), implying that some important facilities may have been left out in the analysis. Especially in countries where Google maps does not provide a comprehensive overview of all services, like LMICs. Did the authors consider making a comparison between their facility list and master facility lists for some countries that are publicly available? If so, what were the results? I would highly recommend that especially for countries in LMICs the authors do some cross-validation checks with governmental facility data.

Response: Thank you for the question. We indeed compared with the CLUES catalog for Mexico (http://www.dgis.salud.gob.mx/contenidos/intercambio/clues_gobmx.html). Unfortunately, mapping between facility datasets is challenging, with many facility lists lacking consistent geographical coordinates or street address information (e.g., some are based on directions and word-of-mouth).

Another reason for using this Google dataset is that this study involves measuring revealed accessibility via anonymized trip travel time information. As such, we needed to ensure consistency between the method of defining trips and the facility location information.

We have added to Methods:

“...this dataset was used to ensure consistency with anonymized trip location information, described below.”

In addition, we have cited South et al (now ref 43) alongside Hierink et al. (now ref 33).

Limitations has also been updated to say “, so additional countries will need to be covered in future work as smartphone ownership and the coverage of healthcare facilities increases.”

R4.8: 8. The authors decide to have 193 countries with potential travel time and 120 countries with revealed travel time. I would highly recommend making the comparison only between countries for which you have both revealed and potential travel time and thus reducing the total number to 120. I would expect that for many countries with potential travel time that do not have revealed travel time, the actual travel time may be much larger in rural areas, where at the moment more traditional friction layers may be more representative of reality than the Google API is now known to be.

Response: This is an excellent point. Whereas indeed the full revealed (120) country list is presented in Figure 1, Figure 6 that does a head-to-head revealed vs. potential comparison only includes the overlapping countries. To make this clearer, we have included the following sentence at the end of the potential accessibility Results section: “A direct comparison between potential and revealed accessibility (in the same countries) is described next.”

R4.9: 9. In the discussion, the authors state: “potential travel times fail to fully capture real-world conditions reflecting vehicle ownership; facility closure; and utilization of more distant facilities due to specialized care, insurance considerations, or individual preference”. Yet, revealed travel time also fails to capture these factors. A Google user might be a taxi driver. And if revealed travel time manages to capture facility closure, how did the authors identify closed facilities? Did they assume facilities were closed when they could see the user by-passed a more near facility? This currently remains unclear in the text and should be addressed.

Response: We appreciate this detailed discussion. First, we edited the highlighted statement to be clearer about what we mean (that potential accessibility does not account for those factors at all, whereas the actual trips taken incorporate user decisions that in turn incorporate those factors). The new text reads as follows:

“Compared to revealed measurements, potential travel times fail to fully capture consequences of real-world factors such as vehicle ownership; facility closure; and utilization of more distant facilities due to specialized care, insurance considerations, or individual preference”

Regarding the points around non-healthcare trips (taxi driver), the impact of this is discounted by virtue of this trip counting at most once since every user is only represented at most once (please refer to the second paragraph of the Methods section “Revealed Accessibility”). More generally, visitors may also be counted by this analysis, as described in the limitations section: “Second, to preserve user privacy, we used a radius of 500m around each facility, which potentially included some non-healthcare-seeking trips in the analysis (the analysis may also include trips by people who visit hospital patients but do not receive healthcare themselves).”

Because trips were anonymized via aggregation, we did not capture the exact times of each trip and whether the destination facility was open at that time. However, if a facility was closed and the person leaves immediately after (or continues on to a different facility), the clustering method behind trip definitions would have defined that closed facility as an intermediate stop instead of the end of a trip (please see “A.1 Computation of trips from raw location data” in the supplement). We have added to the limitations to clarify that closed facility analysis was not accounted for explicitly:

“Furthermore, patients may choose to receive healthcare not in the nearest facility, if a different facility is covered by insurance or was recommended by the referring physician or a family member, if the nearest facility is closed at that time, or if care is delivered in the home or in a community setting.”

R4.10: 10. The authors used aggregated measures of LandScan population data at 5 km². I’m wondering why the authors decided to go for LandScan when more refined estimates are available from other data sources and differences between population datasets are found to be large (Hierink et al., 2022). WorldPop refined the population estimates in their constrained layers by using building footprint data. By using building footprint data, the population estimates are constrained to places where actual settlements are detected, minimizing the possibility of having population counts in extremely uninhabitable and unpopulated places. In addition, why did the authors decide to aggregate this data and not calculate the nearest facility for each population point/grid cell and then take the median or average of all these distances and travel times? I encourage the authors to elaborate on their decision-making behind this and highly advise them to consider another data source and analysis method for the population aggregation.

Response: Thank you for the pointer and the helpful reference. Hierink et al. state in their paper: “Coverage statistics are highest when using GHS-POP (95.0%), HRSL (93.2%), LandScan (93.4%), or WorldPop top–down constrained (84.3%).” Given this and the fact that travel times in their study (Figure 1 in the paper) were very divergent but converged at the higher end, it’s likely that differences narrow for the longer travel times analysis.

We have added to the Methods: “Whereas Landscan has one of the highest population dataset coverage 33, exploration of other population datasets may be helpful.”

We have also revised the Methods to be clearer that LandScan was used specifically for potential accessibility and not revealed accessibility: “For potential accessibility computation, to identify the location of the nearest medical facility for all populated areas (with a population of at least 50 people based on Landscan data),...”

Regarding the suggestion on using an even more fine-grained grid, we agree that this would be more granular, but substantially more computationally expensive to compute directions for. In addition, using a finer grid would have led to more cells having fewer than 50 people. That said, we agree that in principle this could be explored in more detail and have edited the above Methods statement to say “...exploration of other population datasets and geographical granularities may be helpful.”

R4.11: 11. In the manuscript the authors state: “Conversely, substantially higher potential travel times to healthcare facilities by car were observed in sub-Saharan Africa.” Yet only a very small number of sub-Saharan countries were included in the study. At the moment the study lacks a strong foundation on how the authors made sure that the data they had for LMIC countries was representable and why in this case they did not consider transport by foot. The authors do explain that smartphone and car ownership is likely to be biased towards individuals with higher incomes and livelihoods. This is probably not representable of the most vulnerable. The authors need to strengthen this aspect throughout the manuscript. Including as well that the impact of COVID-19 and the measures taken were very different between countries. This aspect is not emphasized at the moment.

Response: We appreciate this comment and have updated the mentions of global to ensure we are clearer that this analysis does not cover all countries, but rather more than 100 countries.

The point on walking is a good one, and the walking data is presented in Figure A.3. We have added to the caption “Data for other modes of transportation are presented in Figure A.3.” to make this clearer.

R4.12: 12. The legend of Figure 3 should also included a symbol for the grey countries.

Response: Thank you, we have added: “Gray color indicates countries or regions not included in the study”

Reviewers' comments:

Reviewer #1 (Remarks to the Author):

I would like to thank the authors for submitting the revised manuscript which reads much better. I would recommend publication of the manuscript.

Reviewer #2 (Remarks to the Author):

The authors have addressed each of the comments and questions I raised during my first review - thank you for the detail and responses in the rebuttal. I think the addition of the colour-coding of figure 1 to show WB income classification is a useful addition. I do still think the inclusion of names of all countries in figure 4 is a challenge, but apart from only including the names for outlier countries, I agree that there are limited options for the figure in its current form. Aside from these points, I have no significant comments or suggestions to make on the revised manuscript. Well done.

Reviewer #3 (Remarks to the Author):

None

Reviewer #4 (Remarks to the Author):

I thank the authors for the extensive revisions made. They have addressed all my concerns sufficiently. I would only highly recommend them to revisit the changes made in response to answer R4.10. The fact that landscan has one of the highest coverages does not mean or guarantee that it is the correct population distribution. The authors in Hierink et al. (2022) show that very large sub-national variations persist and recommend that for the purpose of accessibility analyses HRSL or WorldPop constrained may be the most useful because of their resolution and the precision of the building footprint used in the data generation. The authors should really reconsider the statement made and think about updating their estimates based on one of these layers. Figure 1 in Hierink et al. (2022) shows these differences at the continental level so naturally the differences will smooth out towards the higher travel times. But when looking at the sub-national maps, the large discrepancies between the datasets can be seen.

Response to reviewers

We thank the reviewers for their thoughtful reviews and acknowledging our addressing of their comments! We address the only remaining critique (the comment from Reviewer 4) below in blue.

Reviewer #1 (Remarks to the Author):

I would like to thank the authors for submitting the revised manuscript which reads much better. I would recommend publication of the manuscript.

Response: We appreciate the reviewer's careful review and positive recommendation.

Reviewer #2 (Remarks to the Author):

The authors have addressed each of the comments and questions I raised during my first review - thank you for the detail and responses in the rebuttal. I think the addition of the colour-coding of figure 1 to show WB income classification is a useful addition. I do still think the inclusion of names of all countries in figure 4 is a challenge, but apart from only including the names for outlier countries, I agree that there are limited options for the figure in its current form. Aside from these points, I have no significant comments or suggestions to make on the revised manuscript. Well done.

Response: We appreciate the reviewer's careful review and positive recommendation.

Reviewer #3 (Remarks to the Author):

None

Response: We appreciate the reviewer's careful review.

Reviewer #4 (Remarks to the Author):

I thank the authors for the extensive revisions made. They have addressed all my concerns sufficiently. I would only highly recommend them to revisit the changes made in response to answer R4.10. The fact that landscan has one of the highest coverages does not mean or guarantee that it is the correct population distribution. The authors in Hierink et al. (2022) show that very large sub-national variations persist and recommend that for the purpose of accessibility analyses HRSL or WorldPop constrained may be the most useful because of their resolution and the precision of the building footprint used in the data generation. The authors should really reconsider the statement made and think about updating their estimates based on one of these layers. Figure 1 in Hierink et al. (2022) shows these differences at the continental level so naturally the differences will smooth out towards the higher travel times. But when looking at the sub-national maps, the large discrepancies between the datasets can be seen.

Response: We appreciate the reviewer's careful consideration of our manuscript and suggestion to incorporate the insights from Hierink et al. We copy the previous round's comment below for ease of reference:

“R4.10: 10. The authors used aggregated measures of LandScan population data at 5 km². I’m wondering why the authors decided to go for LandScan when more refined estimates are available from other data sources and differences between population datasets are found to be large (Hierink et al., 2022). WorldPop refined the population estimates in their constrained layers by using building footprint data. By using building footprint data, the population estimates are constrained to places where actual settlements are detected, minimizing the possibility of having population counts in extremely uninhabitable and unpopulated places. In addition, why did the authors decide to aggregate this data and not calculate the nearest facility for each population point/grid cell and then take the median or average of all these distances and travel times? *I encourage the authors to elaborate on their decision-making behind this and highly advise them to consider another data source and analysis method for the population aggregation.*”

We appreciate the opportunity to elaborate on our decision making process.

The novel aspect of our work lies in extracting revealed travel times (as measured by actual human movements) to healthcare facilities and contrasting those measurements with ‘potential’ travel times (based on patterns of private and public transport). The reviewer’s comment contains two critiques: one relating to the methodological approach and another relating to the choice of population surface data for populating weighting when producing national-level estimates. It is important to emphasize that both critiques relate only to the potential travel times component of this work.

For the methodological critique, our approach did derive all possible point-to-facility travel times using the Google Maps API, which we then compared to measured (i.e., revealed) travel times. We then aggregated these differences to the country level before and during the COVID pandemic to show the summarized differences.

For the choice of population surface, our potential accessibility data were computed in mid-2019 and then repeated in mid-2021. The former year was necessary and appropriate for this work because our baseline periods for COVID were in 2019 (i.e., pre-COVID). Please note that the Hierink work was not published at that point and could not be incorporated into our analysis plans.

At this time, there is no way to re-compute historical potential accessibility data (based on 2019 or 2021); (i.e., Google does not have an API that computes travel directions and time estimates as of a past day/time). Consequently, using the API today with an alternate population surface would only provide potential travel time estimates as of 2023. Thus, we cannot construct an analysis that would recreate our work in LandScan with another dataset today and would lose the COVID movement analysis component of this research by switching population surfaces at this point.

Hierink et al. state in their paper: “Coverage statistics are highest when using GHS-POP (95.0%), HRSL (93.2%), LandScan (93.4%), or WorldPop top–down constrained (84.3%).” Given this and the fact that travel times in their study (Figure 1 in the paper) diverged for different population surface datasets but converged at the higher end, it’s likely that differences diminish for the longer travel times analysis. Furthermore, we agree that the most recent iteration of WorldPop is superior to the LandScan data used in our analysis, but no gridded population surfaces are perfect representations of reality for reasons that include their reliance on rarely conducted national censuses and the fact that humans regularly move beyond their home location on multiple time scales.

Thus, we would like to propose the following steps (1-3 have been done and acknowledged by the reviewer; 4 and 5 are new) to address the concerns of Reviewer 4:

1. We have added the following sentence to the Methods section: “Whereas Landscan has one of the highest population dataset coverage^{33,43}, exploration of other population datasets may be helpful.”
2. We have also revised the Methods section to be clearer that LandScan was used specifically for potential accessibility and not revealed accessibility: “For potential accessibility computation, to identify the location of the nearest medical facility for all populated areas (with a population of at least 50 people based on Landscan data),...”
3. Regarding the suggestion on using an even more fine-grained grid, we agree that this would be more granular, but would have led to more cells having fewer than 50 people. For completeness and to reflect the reviewer’s suggestion, we have edited the above Methods statement to say “...exploration of other population datasets and geographical granularities may be helpful.”
4. Revising the relevant Methods part (bold is new) as follows: “Whereas Landscan has one of the highest population dataset coverage,^{33,43} exploration of other population datasets **such as HRSL or WorldPop** and geographical granularities may be helpful, **particularly in future work focusing on sub-Saharan Africa where subnational data differ substantially between datasets³³. Implementing this analysis framework on a longitudinal basis would provide the most robust results if a single population dataset is selected, and ideally a dataset that is deemed to best fit the population distribution of the specific target country or region.**
5. We provide below a figure comparing the Landscan 2017 data with WorldPop 2020 across approximately 190 countries and would be happy to incorporate insights from this figure (eg, maximum country-level deviation) into the paper if helpful.

LandScan vs WorldPop Country Population (excluding CN & IN)

Legend: Recent studies suggest that differences exist between various population data sets in analyzing potential travel times for sub-saharan Africa (reference Hierink et al). We compared the LandScan 2017 data with WorldPop 2020 across approximately 190 countries. The national level population counts were highly correlated; with differences (ranging from 0% to approximately 25%) possibly related to population growth due to the three year time gap. These results are consistent with other analyses using LandScan and WorldPop as a population data source (reference Hierink et al).

We hope that our proposed changes address the remaining concerns of Reviewer 4.

REVIEWERS' COMMENTS:

None